# PI3K/Akt1 signalling specifies foregut precursors by generating regionalized extra-cellular matrix

S Nahuel Villegas[1,2‡], Michaela Rothová[2†], Martin E Barrios-Llerena[3†], Maria Pulina[4,5], Anna-Katerina Hadjantonakis[4], Thierry Le Bihan[3], Sophie Astrof[5], Joshua M Brickman[1,2*]

[1]Institute for Stem Cell Research, MRC Centre for Regenerative Medicine, University of Edinburgh, Edinburgh, United Kingdom; [2]The Danish Stem Cell Center (DanStem), University of Copenhagen, Copenhagen, Denmark; [3]Centre for Synthetic and Systems Biology (SynthSys), University of Edinburgh, Edinburgh, United Kingdom; [4]Developmental Biology Program, Sloan-Kettering Institute, New York, United States; [5]Center for Translational Medicine, Jefferson Medical College, Philadelphia, United States

**Abstract** During embryonic development signalling pathways act repeatedly in different contexts to pattern the emerging germ layers. Understanding how these different responses are regulated is a central question for developmental biology. In this study, we used mouse embryonic stem cell (mESC) differentiation to uncover a new mechanism for PI3K signalling that is required for endoderm specification. We found that PI3K signalling promotes the transition from naïve endoderm precursors into committed anterior endoderm. PI3K promoted commitment via an atypical activity that delimited epithelial-to-mesenchymal transition (EMT). Akt1 transduced this activity via modifications to the extracellular matrix (ECM) and appropriate ECM could itself induce anterior endodermal identity in the absence of PI3K signalling. PI3K/Akt1-modified ECM contained low levels of Fibronectin (Fn1) and we found that Fn1 dose was key to specifying anterior endodermal identity in vivo and in vitro. Thus, localized PI3K activity affects ECM composition and ECM in turn patterns the endoderm.

*For correspondence: Joshua.brickman@sund.ku.dk

†These authors contributed equally to this work

‡Present address: Instituto de Neurociencias, Consejo Superior de Investigaciones Científicas-Universidad Miguel Hernández de Elche, Alicante, Spain

**Competing interests:** The authors declare that no competing interests exist.

## Introduction

Understanding the mechanisms regulating axis formation in vertebrate development has been one of the central questions in modern developmental biology. It involves the activity of a number of conserved signal transduction pathways. However, the complex set of morphological rearrangements that occur during gastrulation make it difficult to uncouple the role of specific signals in mediating cell migration from those acting directly on cell specification (*Villegas et al., 2010*). Embryonic stem cell (ESC) differentiation offers a tractable in vitro model, which can be used to complement the analysis of embryos. ESCs are karyotypically normal, self-renewing and pluripotent cell lines derived from the mammalian blastocyst that can be driven to differentiate toward all the three germ layers. Adherent ESC differentiation models, therefore, allow direct access to mechanisms regulating cell fate decisions in a stable defined environment. The inductive activity of specific signalling pathways can be tested on isolated progenitor populations, allowing the deciphering of specific target cells and paracrine interactions (*Murry and Keller, 2008*).

The specification of visceral organs is intricately linked to the establishment of positional identity and begins with the formation of the endoderm germ layer. Endoderm progenitors will give rise to the

**eLife digest** From conception to birth, a single fertilised egg will multiply into trillions of cells, with each cell becoming one of the 200 or so different types of cell that are found in the human body. The development of an embryo is complex and dynamic, with cells giving up their ability to become any cell type and committing to becoming a specific cell type within a given tissue. At the same time, different groups of cells migrate to the appropriate locations within the developing embryo. Although it is challenging to decipher the roles of the individual signalling pathways that control an embryo's development, several important components have been found.

Fibroblast growth factor (FGF) is a protein that regulates the formation of the endoderm: this is the innermost of the three layers of cells that form in the early embryo, and it gives rise to internal organs such as the gut, liver and pancreas. As well as 'telling' cells to become the front part, or anterior, of the endoderm, FGF also controls the migration of these cells within the embryo. However, uncoupling these two roles has been a major challenge, and the molecular mechanisms behind them are unclear.

Now, Villegas et al. have discovered that FGF activates a signalling cascade involving two enzymes called PI3K and Akt1. In lab-grown embryonic stem cells—cells that can be coaxed to become any of the cell types formed during development—this signalling cascade is essential for FGF to trigger differentiation of the cell types found in the anterior endoderm. The PI3K/Akt1 signalling cascade achieves this by reducing the level of a protein called fibronectin in the 'extracellular matrix' that surrounds the cells. This low level of fibronectin will in turn induce cells to stick together in an organized layer; and this rearrangement of cell-cell and cell-matrix interactions appears linked to triggering the differentiation of anterior endoderm cell types.

Villegas et al. showed that the PI3K/Akt1 pathway was also essential for endoderm formation in living mouse embryos. As a normal embryo develops, the anterior endoderm cells move into a 'groove' at the front the embryo, where the level of fibronectin is lower than it is at the posterior end of the embryo.

These findings highlight the importance of the extracellular matrix in the regulation of embryonic development, and should assist in the effort to turn lab-grown stem cells into the useful cell types found in internal organs.

entire digestive track in addition to thymus, thyroid, liver, pancreas, lungs, gallbladder as well as the extra-embryonic component of the visceral yolk sac. Therefore, understanding the basis for endoderm induction is the first step in attempting directed differentiation to organ cell types. In mammals, endoderm induction occurs in two waves. The first occurs at implantation and leads to the formation of the extra-embryonic visceral endoderm and the second occurs during gastrulation. During gastrulation, the precursors of the definitive endoderm emerge from the anterior region of a transient embryonic processing centre known as the primitive streak (PS) (*Zorn and Wells, 2009*). These precursors will either push away, or intercalate into, the visceral endoderm that surrounds the embryo proper (*Kwon et al., 2008*; *Burtscher and Lickert, 2009*). The first endoderm to emerge from the PS region is the anterior definitive endoderm (ADE) or prospective foregut. These cells move forward along the midline of the embryo to convey anterior information to the emerging neural axis and eventually give rise to the liver, pancreas and other derivatives of the foregut (*Zorn and Wells, 2009*). ADE induction is dependent on a network of specific transcription factors downstream of Wnt and high levels of Nodal related TGF-beta signalling. However, as this emerging endodermal signalling centre moves forward during gastrulation, it starts to express antagonists of both these signals, protecting the anterior neural ectoderm from the mesoderm inducing properties of these pathways (*Tam and Loebel, 2007*).

In addition to Nodal and Wnt signalling, we have recently uncovered an additional requirement for Fibroblast Growth Factor (FGF) signalling in endoderm and ADE specification (*Morrison et al., 2008*). FGF signalling is also required to induce the morphological rearrangements that are normally associated with gastrulation and endoderm migration (*Yamaguchi et al., 1994*), including the induction of an epithelial to mesenchymal transition (EMT) that enables the beginning of cell migration. EMT and cell migration are also regulated by the substrate upon which cells adhere, the extra-cellular matrix

(ECM) (**Hynes, 2009**). For example, PS formation can be inhibited on the anterior side of the embryo as a consequence of the visceral endoderm depositing ECM components that block cell migration (**Egea et al., 2008**) and FGF signalling has been linked to visceral endoderm-dependent deposition of basement membrane components in ESC differentiation (**Villegas et al., 2010**). However, while these studies link FGF signalling to ECM composition and cell migration, they do not address how ECM could be involved in patterning the embryonic axis or induction of specific cell types during ESC differentiation.

In this study, we resolve the ability of FGF signalling to induce both endoderm and morphogenetic movements. We uncoupled the signalling downstream of the FGF receptor using an in vitro mESC based model for endoderm induction. We found that endodermal patterning by FGF was dependent on a PI3K/Akt1 activity that temporally restricts EMT. This unexpected PI3K/Akt1 activity, which we show is required both in vivo and in vitro, is transduced via a modification to ECM composition, which in itself, promotes cell type specification. Using liquid chromatography and mass spectrometry (LC-MS), we further identified ECM components and found that Fibronectin (Fn1) levels were an essential determinant of the ECM-dependent endoderm patterning activity. Taken together, our findings link anterior-posterior (A–P) patterning of the endoderm to EMT and uncover a novel ECM-based mechanism for the activity of PI3K/Akt1 signalling during germ layer specification.

## Results

### A specific requirement for PI3K during anterior induction within the endoderm

To address the basis of the requirement for FGF during endoderm differentiation and anterior specification, we applied several small molecule inhibitors to a defined adherent mESC endoderm differentiation model (**Figure 1A**; **Livigni et al., 2009**). Differentiation conditions are shown in **Figure 1A** and the medium used is defined in 'Materials and methods' section. We used BMP4 together with modest doses of the Nodal-like TGF-beta Activin A, to promote the differentiation of mESCs to epiblast-like cells, followed by the further differentiation of epiblast towards PS and endoderm in the presence of FGF and Activin A. Through the use of this staged differentiation protocol we have been able to generate committed ADE from ESCs more efficiently (**Morrison et al., 2008**). We monitored the formation of ADE, anterior primitive streak (APS) and endoderm in general, using either single or combinatorial fluorescent reporter ESC lines generated by gene targeting. These included the fluorescent reporter gene RedStar under the transcriptional control of the early anterior endoderm marker *Hhex* (HRS) (**Figure 1—figure supplement 1A**) and a GFP under the control of the *Goosecoid* (*Gsc*) locus, a marker of PS and APS (**Figure 1—figure supplement 1A,B**). For additional resolution of ESC differentiation to endoderm we used the cell surface marker, Cxcr4 (**Morrison et al., 2008**). ADE was identified as either double positive for Hhex and Cxcr4 (H$^+$C$^+$) or Hhex and Gsc (H$^+$Gsc$^+$). Differentiation of these reporter lines was assessed by quantitative RT-PCR (q-RT-PCR) (**Figure 1—figure supplement 1C**), which confirmed that these cells pass through the in vitro equivalents of specific developmental stages, including PS, APS (or mesendoderm), and ADE, thereby recapitulating in vivo development. Based on these data, we designated the specification phase leading up to PS specification as phase 1 and the stage of subsequent mesoderm and endoderm segregation as phase 2 (**Figure 1A**). The transition between phase 1 and 2 is marked by the appearance of Gsc-GFP cells during day 3.0–3.5 of ESC differentiation.

Application of inhibitors of MEK- (PD0325901–PD032−), JNK (SP600125–SP−), p38 (SB239063–SB−), and PI3K (LY249002–LY−) during phase 2 of differentiation all resulted in an inhibition of ADE specification (**Figure 1C**, **Figure 1—figure supplement 2A–C**). However, while inhibition of different MAPKs (with PD032, SP and SB) also resulted in a dramatic reduction in mesendodermal and pan-endodermal, Hhex$^-$Cxcr4$^+$ (H$^-$C$^+$) populations, only PI3K inhibition with LY had a specific effect on induction of ADE (**Figure 1B,C**). While each of these kinases were required for ADE specification at a certain level, some Hhex$^+$ cells were observed in SP treated cultures, although endodermal gene expression was reduced (**Figure 1—figure supplement 2B**) and these cells co-express the ESC marker Oct4 (**Figure 1—figure supplement 2D**). Thus, all these kinases were required broadly for ESC differentiation towards mesoderm and endoderm, but only PI3K appeared specific to the transition between mesendoderm and committed ADE. To confirm that these signalling requirements were specific to phase 2, we also examined the effects of these inhibitors in phase 1. Inhibition

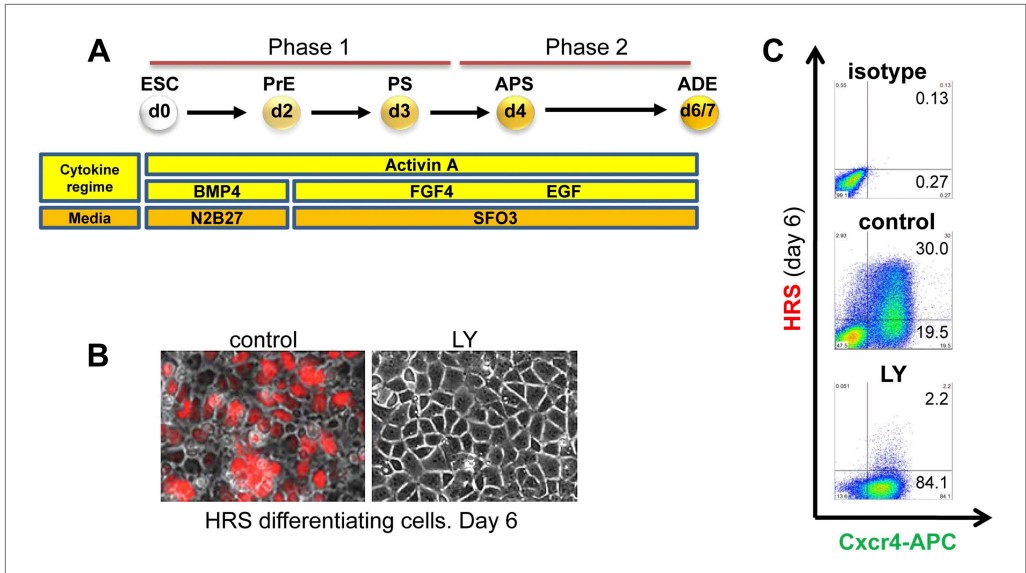

**Figure 1**. Inhibition of PI3K signalling disrupts Hhex positive ADE specification, but does not interfere with mesendoderm induction. (**A**) Schematic representation of ESC differentiation towards ADE. PrE: primitive ectoderm, PS: primitive streak, APS: anterior primitive streak, ADE: anterior definitive endoderm. (**B**) Fluorescence and brightfield images of HRS ADE cultures generated in the absence or presence of LY. LY was present throughout phase 2. (**C**) ADE (H$^+$C$^+$), but not Cxcr4$^+$ mesendoderm (H$^-$C$^+$) was impaired by LY treatment. Gates were set using parental E14 Tg2A (E14) cells without fluorescent reporters. Hereinafter, LY treatment refers 10 μM at d3.5 and 20 μM at d4.5 onwards. For some of the experiments described in this paper the base media SFO3 was substituted by ADEM with identical results.

The following figure supplements are available for figure 1:

**Figure supplement 1**. A model for monitoring endoderm specification from ESCs.

**Figure supplement 2**. MAPK kinase signalling is required for endoderm induction.

of either JNK or PI3K was highly toxic, leading to extensive cell death, even at low concentrations (*Supplementary file 1*). Inhibition of MEK resulted in ESC-like colonies that maintained *Nanog* expression (*Figure 1—figure supplement 2B,E*) consistent with a requirement for MEK signalling during early ESC differentiation (*Kunath et al., 2007*; *Stavridis et al., 2007*; *Ying et al., 2008*). Suppression of p38 signalling with SB also blocked differentiation toward APS derivatives, although SB was not able to support ESC-like phenotypes (*Figure 1—figure supplement 2E*).

Gene expression analysed by q-RT-PCR also indicated that PI3K signalling was essential for anterior endoderm specification. We found that the expression of pan-endodermal markers *Sox17*, *Foxa2* and *Gsc* were enhanced by PI3K inhibition, while induction of all ADE specific gene expression (*Hhex*, *Cer1*, *Lefty1*, *Sfrp5* and *Fzd5*) was repressed (*Figure 2A*). The loss of Hhex expression in the absence of PI3K signalling within the Sox17$^+$/Foxa2$^+$ progenitor population was confirmed by immunocytochemistry (IC) (*Figure 2B* and data not shown). Taken together these findings suggest that PI3K is required after PS stages for the acquisition of anterior positional identity in the endoderm and, when it is suppressed, cells acquire an earlier more naïve or potentially posterior endodermal state.

We confirmed that PI3K was active during normal endoderm differentiation based on the phosphorylation of its main target, Akt1 (pAkt1). pAkt1 was down-regulated just as cells enter phase 1, but levels were rapidly restored at day 2 of differentiation, and maintained throughout the endodermal differentiation (*Figure 2C*, upper panel). LY rapidly suppressed Akt1 phosphorylation and while pAkt1 levels exhibited a slight recovery 6 hr after LY addition, they remained low throughout differentiation (*Figure 2C*, middle and bottom panel).

As phosphorylation of Akt1 is required for cell survival, we asked if the phenotype of LY was driven by selective death of ADE precursor cells. Doses of LY that completely suppressed pAkt1 (50 μM)

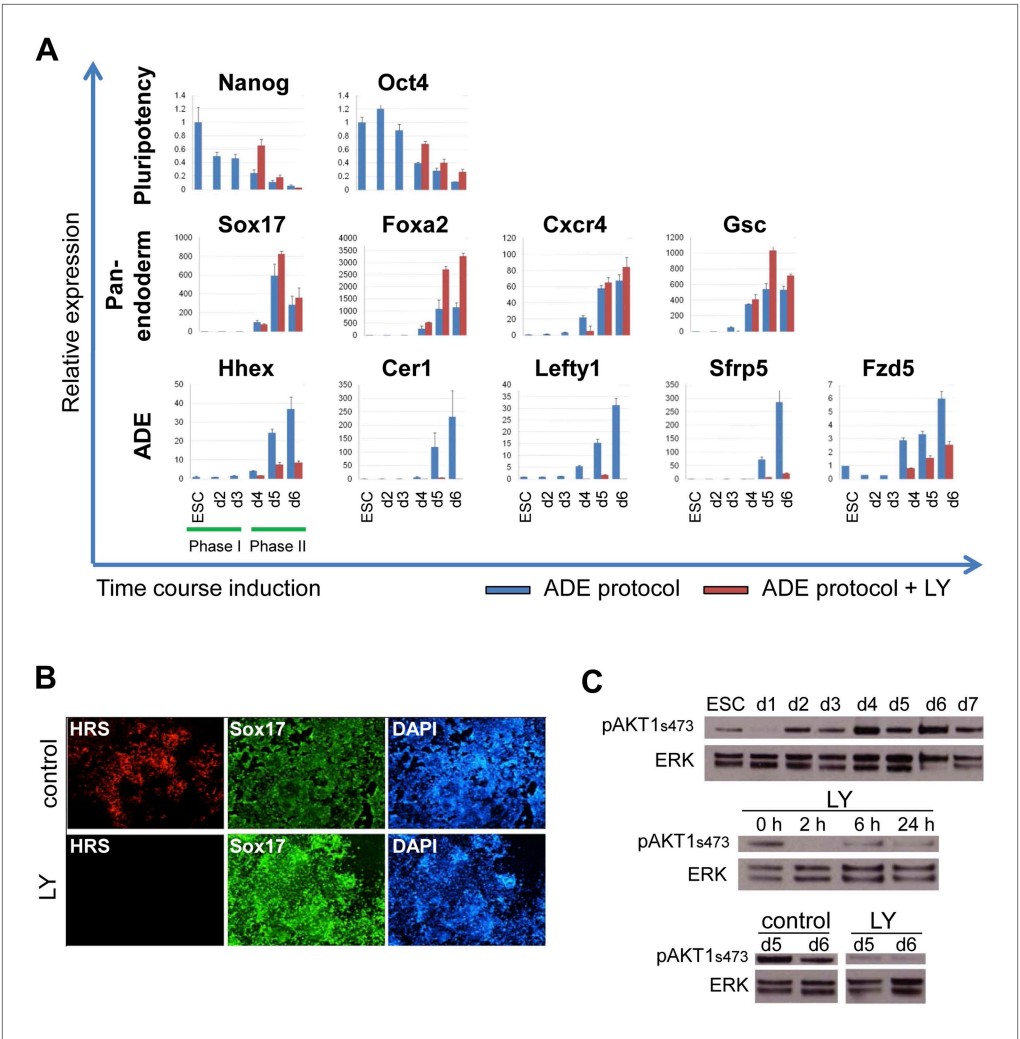

**Figure 2**. Inhibition of PI3K signalling inhibits ADE but not DE differentiation. (**A**) Differential regulation of ADE, pan-endoderm and pluripotency markers as a result of LY treatment. Q-RT-PCR showing relative gene expression for ESC, endodermal and ADE markers in ESC differentiation towards endoderm. Transcript levels were normalised to the *Tbp* value obtained for each sample. Normalised values are shown relative to ESC expression level. ESCs were differentiated to ADE using the protocol described in *Figure 1* in the presence or absence of LY. (**B**) Treatment with LY impaired HRS but not Sox17 expression. Fluorescence images showing HRS expression and Sox17 antibody staining at day 6 of differentiation. (**C**) Phosphorylation of Akt1 in ESC differentiation. Western blot showing phosphorylation of Akt1 during a time course for endoderm differentiation toward ADE (upper panel) and its dephosphorylation after the addition of 20 µM LY at different time points (middle panel, h; hours, top and bottom panel, d; days). Total ERK was used as loading control.
The following figure supplements are available for figure 2:

**Figure supplement 1**. PI3K is not specifically required for ADE survival.

resulted in extensive cell death (*Figure 2*, *Figure 2—figure supplement 1B*, *Supplementary file 1*), while medium doses (20 µM) lead to a transient increase in apoptosis in Gsc⁻, but not Gsc⁺ populations of day 3 cultures as judged by Annexin V staining (*Figure 2*, *Figure 2—figure supplement 1A,C,D*). However, by day 4, LY had no effect on survival of either Gsc⁻ or Gsc⁺ cells (*Figure 2—figure supplement 1C,D*). To further confirm that LY was not blocking ADE induction through the selective cell death of a precursor population, we inhibited apoptosis with Z-VAD-FMK, a specific caspase inhibitor. The addition of this inhibitor to LY containing cultures eliminated the transient increase in

apoptosis observed in the Gsc⁻ population, but produced no rescue of the ADE induction (*Figure 2*, *Figure 2—figure supplement 1B*).

## PI3K inhibition prevents endoderm from exiting a naïve endoderm state

We found that PI3K was required early during endoderm specification for the eventual induction of anterior cell types. *Figure 3A* shows the induction of ADE at day 7, in cultures exposed to LY for different periods of time, and demonstrates that PI3K was essential between days 3 and 5 of differentiation. Thus, the induction of ADE was dependent on the action of PI3K early during phase 2. During this phase of differentiation two populations of Gsc-GFP expressing cells (Gsc^low^ and Gsc^high^) become apparent. We found that Hhex⁺ ADE cells were always Gsc^high^ suggesting they arose specifically from the earlier Gsc^high^Hhex⁻ population. We also found that this Gsc^high^ population was diminished in the presence of LY while the Gsc^low^ population was considerably expanded (*Figure 3—figure supplement 1A,B*). Thus, PI3K was required in early PS precursors to generate Gsc^high^ cells that can later give rise to Hhex⁺ ADE. When LY-treated cells were released from the PI3K block and cultured for an additional 3 days in phase 2 differentiation conditions, we observed a complete recovery of Hhex induction, suggesting that in the absence of PI3K cells fail to progress in differentiation, but they were maintained in a less differentiated endodermal naïve state (*Figure 3B*). To better characterize this LY-dependent naïve

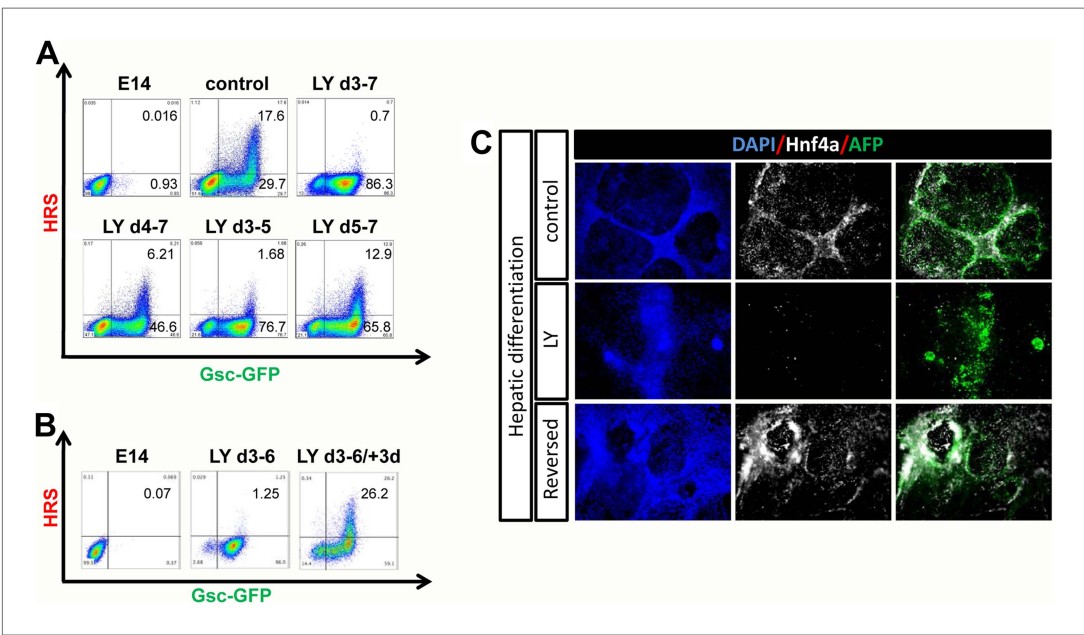

**Figure 3**. PI3K promotes exit from a naïve endoderm state. (**A**) PI3K is essential during endoderm segregation. Time course for endoderm differentiation analysed by flow cytometry on the HRS/Gsc-GFP cells showing a requirement for PI3K signalling during days 3–5 of ADE differentiation. Cells were differentiated under normal conditions or treated with LY for different periods of time. The period of LY treatment is stated for each graph and ADE induction was assessed at day 7 of differentiation. Gates were set with parental E14s as in *Figure 1*. (**B**) Cells exposed to LY during d3–6 could be returned to normal differentiation to generate ADE. Flow cytometry showing the emergence of the Hhex⁺/Gsc⁺ ADE after LY-treated naïve endoderm was returned to normal differentiation for a further 3 days. Gates were set as in **A**. (**C**) Hepatic differentiation of ADE and naïve endoderm. Immunostaining for AFP and Hnf4a on differentiated cells showing the formation of hepatocyte progenitors. ADE (control-upper panel), cells differentiated in LY (middle panel) and ADE generated from LY-treated naïve cells (lower panel) were subjected to hepatocyte differentiation. LY-treated cells showed reduced differentiation efficiency, whereas both ADE and recovered LY-treated ADE efficiently generated hepatocytes.

The following figure supplements are available for figure 3:

**Figure supplement 1**. PI3K inhibition supports a naïve endoderm state.

state we assessed a panel of markers by q-RT-PCR and IC. Alongside the high levels of endodermal markers previously observed (*Sox17, Foxa2*) in LY-treated cultures, they also showed expression of early primitive streak markers such as *Mixl1* and *Cdh2*, but not mesodermal specific gene expression (*Meox1, Meox2, Mesp1, Mesp2, Isl1, Gata1*) (*Figure 3—figure supplement 1C* and data not shown). While these cultures expressed high level of PS gene expression, this appears to be predominantly endoderm as 90% of the Gsc⁺ cells in these LY-treated cultures co-expressed Sox17 (*Figure 3—figure supplement 1D,E*). These cells also do not up-regulate the expression of posterior (*Cdx2*), or visceral (*Hnf4a, Dab2*) endoderm markers. Taken together, these data indicate that blocking PI3K signalling retains endodermal cells in a naïve state.

To further confirm that the LY-mediated block to differentiation maintained a true functional naïve state, we tested the capacity of these cells for further differentiation towards hepatocyte or Pdx1 positive pancreatic progenitors (*Morrison et al., 2008*). While ADE cells progressed efficiently into both lineages, LY-treated cells failed to differentiate efficiently (*Figure 3C*, *Figure 3—figure supplement 1F*). However, when LY-treated naïve cells were returned to normal differentiation for 3 days prior to subjecting them to hepatocyte differentiation, they performed as normal Hhex positive ADE cultures, efficiently generating Hnf4a/AFP positive hepatocyte progenitors (*Figure 3C*). Endodermal cultures were also subjected to intestinal differentiation (*Spence et al., 2011*), but we were unable to generate intestinal spheres from either ADE or LY-treated cells (data not shown). This may indicate that specific posterior pre-patterning of naïve endoderm is required. Taken together, these experiments suggest that the cells produced by PI3K/Akt1 inhibition are non-regionalized endoderm (naïve) that retains the ability to give rise to specific endodermal domains when returned to normal differentiation.

## PI3K acts to enhance and maintain epithelialization in the forming endoderm

We found that the Gsc^low population generated in the presence of LY failed to make cell–cell contacts and no longer maintained the epithelial morphology normally observed at this early stage of endoderm differentiation (*Figure 3—figure supplement 1B,D*, bright field). Importantly, normal up-regulation of E-cadherin was not observed and residual E-cadherin was not localized to cell–cell junctions, while the pro-EMT transcription factor Snai1 was up-regulated (*Figure 4A,B*). These observations indicated that in the absence of PI3K signalling, naïve endoderm precursor cells remained in an extended EMT-like process, instead of forming an organized epithelium.

Since differentiating populations of ESC showed heterogeneous expression of Gsc, we investigated whether Akt1 was preferentially activated in a particular population. *Figure 4C* shows that pAkt1 levels were significantly higher in the Gsc^high cells sorted from differentiating ESCs at the time at which Gsc^high and Gsc^low states could first be distinguished (day 4.5). Q-RT-PCR analysis on the different fractions also indicated that the Gsc^high pAkt1^high population was endodermal (*Sox17*⁺), whereas the Gsc^low pAkt1^low population formed under control conditions was still mesendodermal (*Brachyury*⁺) (*T*⁺) (*Figure 4D*). As the Gsc^high pAkt1^high population is lost upon LY addition, this indicates that anterior endoderm generation would appear to require the activation of Akt1. The Gsc^low population formed in the presence of LY was also distinct from the Gsc^low population formed under control conditions. Gsc^low-LY treated cells expressed high levels of *Sox17* whereas *T* expression was reduced (*Figure 4E*). Thus, while LY may promote an EMT-like state, it is not promoting mesodermal, but rather naïve mesenchyme-like endodermal state.

## Akt1 activation is sufficient to induce anterior endodermal identity

We uncoupled Akt1 activation from PI3K signalling by using an Akt1 fusion to the oestrogen receptor, Myr-Akt1-mER (*Kohn et al., 1998*), that placed activated and myristoylated Akt1 under the control of the oestrogen analogue 4-hydroxy-tamoxifen (Tam) (*Figure 5—figure supplement 1A,B*). The Myr-Akt1-mER fusion protein was constitutively induced in HRS cells and its expression was visualized based on GFP expression from an internal ribosomal entry site (IRES) (Myr-Akt1-mER-IRES-GFP/HRS) (Akt1-GFP-HRS) (*Figure 5—figure supplement 1A–C*). In this cell line, we found that Tam stimulated Akt1 activation rescued ADE generation in the presence of LY. This ability of pAkt1 to support ADE specification was observed both by flow cytometry (*Figure 5A*) and by q-RT-PCR (*Figure 5C*). ADE markers (*Hhex, Sfrp5, Fzd5*) were induced robustly in response to Tam, alongside the rescue of ADE epithelial morphology, even in the presence of LY (*Figure 5B*).

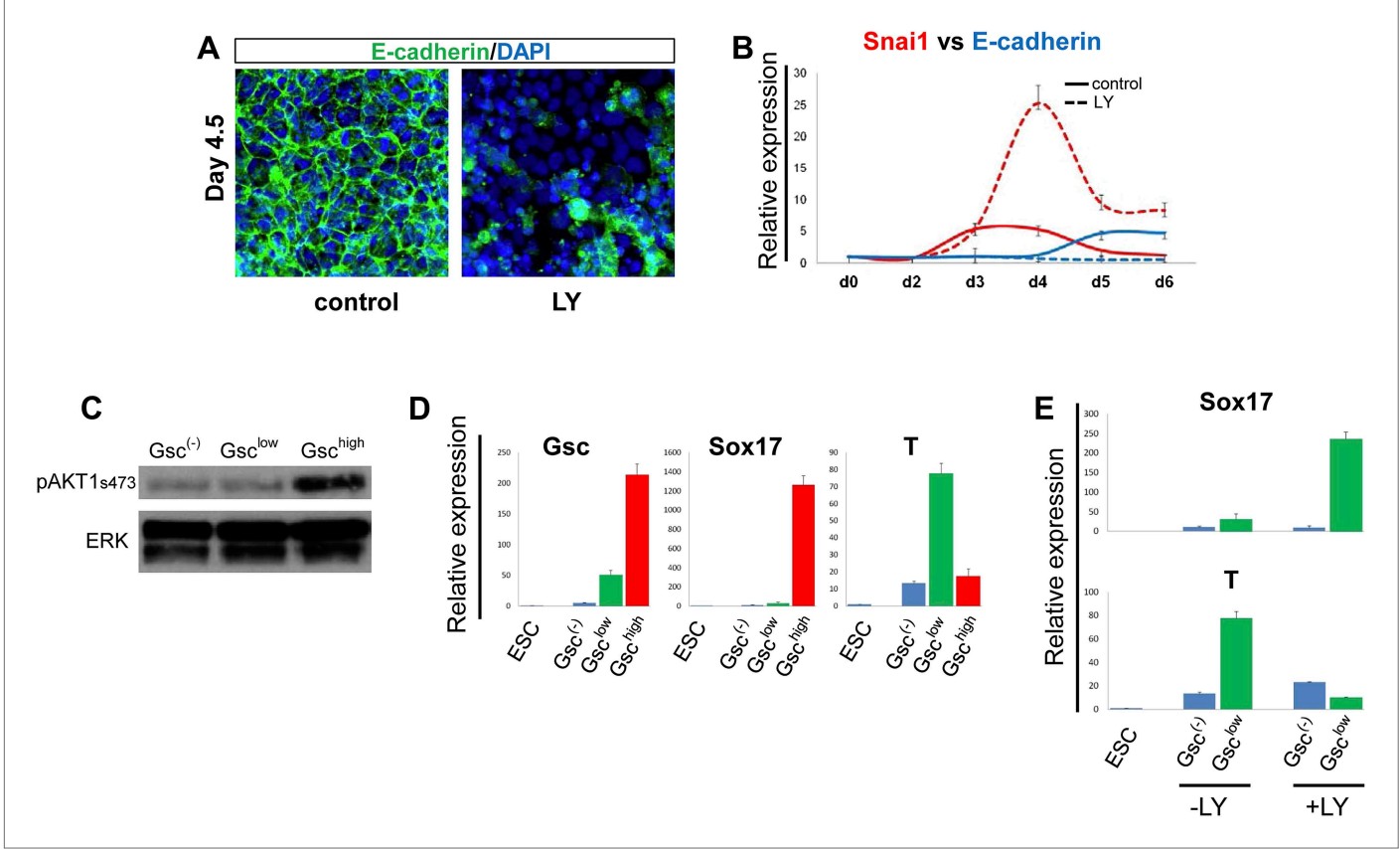

**Figure 4**. PI3K/Akt1 regulates EMT and is localized to the pre-ADE population. (**A**) PI3K activity is necessary to maintain epithelial integrity in endoderm differentiation. Immunofluorescense images showing strong expression of the epithelial marker E-cadherin during ADE specification. Addition of LY during phase 2 resulted in a reduction of E-cadherin levels and disappearance of E-cadherin from cell–cell junctions. (**B**) Q-RT-PCR showing a failure in *E-cadherin* transcriptional up-regulation as well as the up-regulation of its repressor and EMT inducer, *Snai1*, in LY-treated cultures. Transcript levels were normalised as described in *Figure 2A*. Control: ADE differentiation protocol. (**C** and **D**) Normal differentiating HRS/Gsc-GFP cells were sorted by flow cytometry based on *Gsc* levels at day 4.5 of ADE differentiation. (**C**) Western blots showing Akt1 phosphorylation in different fractions of differentiating ESC. pAkt1 was localized to the Gsc^high population. Total ERK was used as loading control. (**D**) Q-RT-PCR showing lineage markers in sorted cells. The Gsc^high/Akt1^high population coincides with the *Sox17* positive emergent endoderm. (**E**) Inhibition of PI3K during phase 2 alters the identity of the Gsc^low population. Q-RT-PCR showing the mesoderm and endoderm markers *T* and *Sox17*. Transcript levels for **D** and **E** were normalised as described in *Figure 2A*.

## Akt1 signalling acts cell non-autonomously to specify anterior endoderm

As the Akt1-GFP-HRS cells contained a constitutive GFP reporter, we mixed Akt1-GFP-HRS expressing cells with unmodified HRS ESCs, and assessed the induction of endoderm in the presence of LY, in both the Akt1 expressing and non-expressing populations (*Figure 5—figure supplement 1D*). We found that the stimulation of Akt1 by Tam administration rescued the block to ADE induction in both Akt1 expressing and non-expressing cells (*Figure 5D*, *Figure 5—figure supplement 1E*), indicating cell autonomous and non-autonomous rescue of the LY phenotype. However, based on a time course for induction of gene expression in sorted Akt1-GFP⁺ and Akt1-GFP⁻ populations (+/− Ly, +/− Tam), it appeared that Akt1 also had cell autonomous effects on *Snai1* and *Sox17* transcription (*Figure 6—figure supplement 1A*).

## pAkt1 supports ADE induction via the generation of a specific ECM

To identify the factors downstream of pAkt1 that mediate non-autonomous ADE induction, we tested the ability of supernatants produced during normal differentiation to rescue ADE generation in LY-treated cultures, but failed to observe any effect (data not shown), suggesting that these

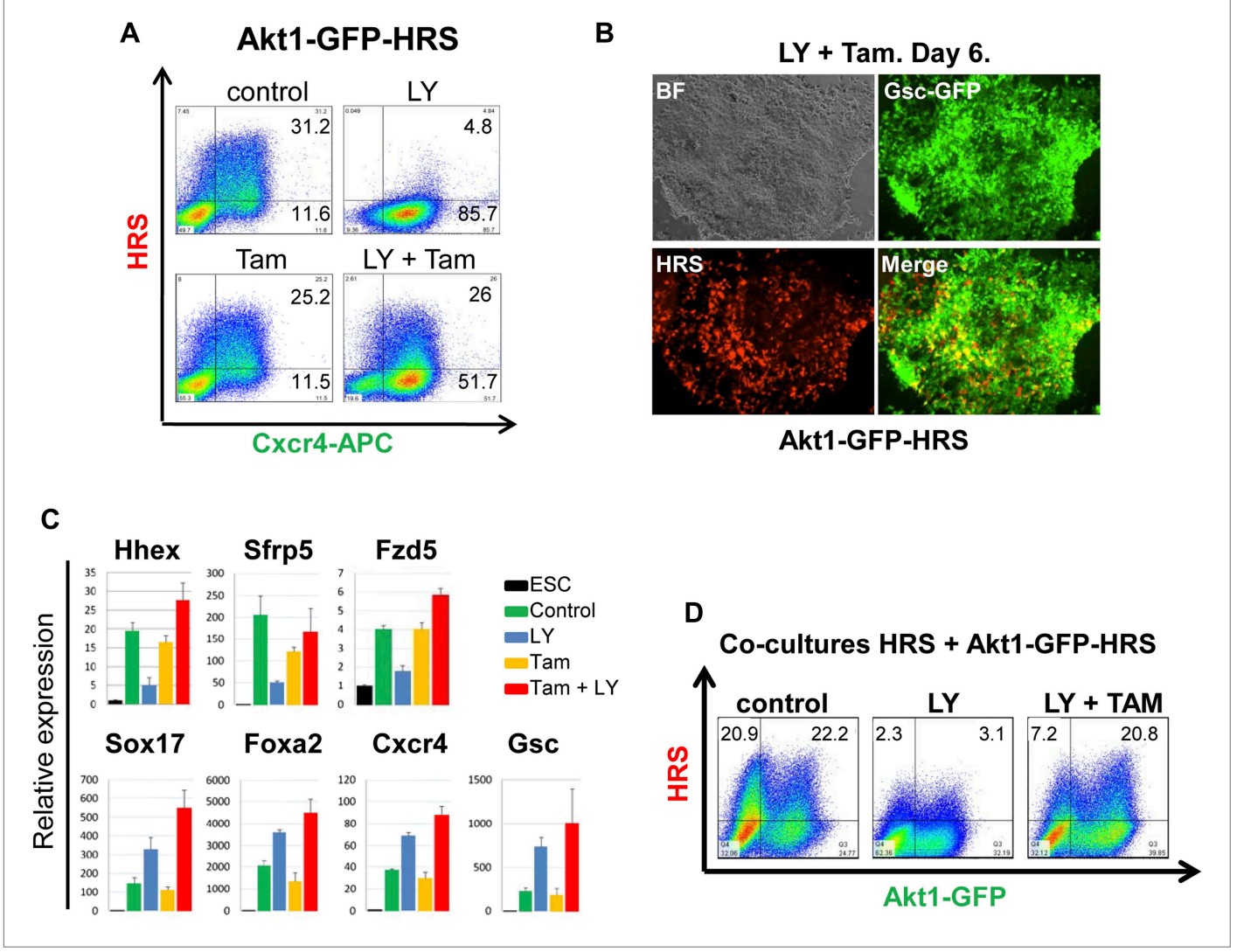

**Figure 5**. Akt1 activation rescues the block to PI3K in ADE differentiation. (**A**) ADE induction is rescued by Akt1 activation. Flow cytometry on Akt1-GFP-HRS differentiating cells showing that induction of Akt1 with Tamoxifen (Tam) rescues HRS expression. (**B**) Fluorescence microscopy showing Akt1-GFP-HRS differentiating cultures in presence of LY and Tam. HRS expression and ADE epithelial morphology were both rescued. (**C**) Q-RT-PCR showing rescue of ADE marker expression in LY-treated cultures by Tam-induced Akt1 expression. Transcript levels were normalised as described in *Figure 2A*. (**D**) Flow cytometry showing cell non-autonomous rescue of mixed Akt1-GFP-HRS and HRS differentiating co-cultures treated with Tam and/or LY. Akt1-GFP-HRS cells can be distinguished based on GFP expression (X-axis). HRS expression (Y-axis) was rescued in both Akt1-GFP-HRS expressing and non-expressing cells.

The following figure supplements are available for figure 5:

**Figure supplement 1**. Activation of Akt1 supports Hhex induction in the presence of a block to PI3K.

factors might not readily diffuse. We therefore tested the hypothesis that the ADE inducing activity downstream of Akt1 might be the result of specific Akt1-dependent ECM proteins. We reasoned that if the role of Akt1 in anterior patterning is conducted by the production of a specific ECM, then if we expose differentiating cells to the action of an ECM generated under normal conditions, pAkt1 would no longer be necessary for ADE induction. To do so, we prepared ECM from untreated differentiating ESC cultures (ECM1) or from cultures treated with LY (ECM2). These ECM preparations were then tested for their ability to induce or rescue endoderm differentiation in the presence of LY (*Figure 6B*). HRS-Gsc-GFP cells were differentiated to the APS GFP+ stage (*Figure 1A*), collected, re-plated onto the different matrices or gelatine (*Figure 6A*) and differentiated in the presence

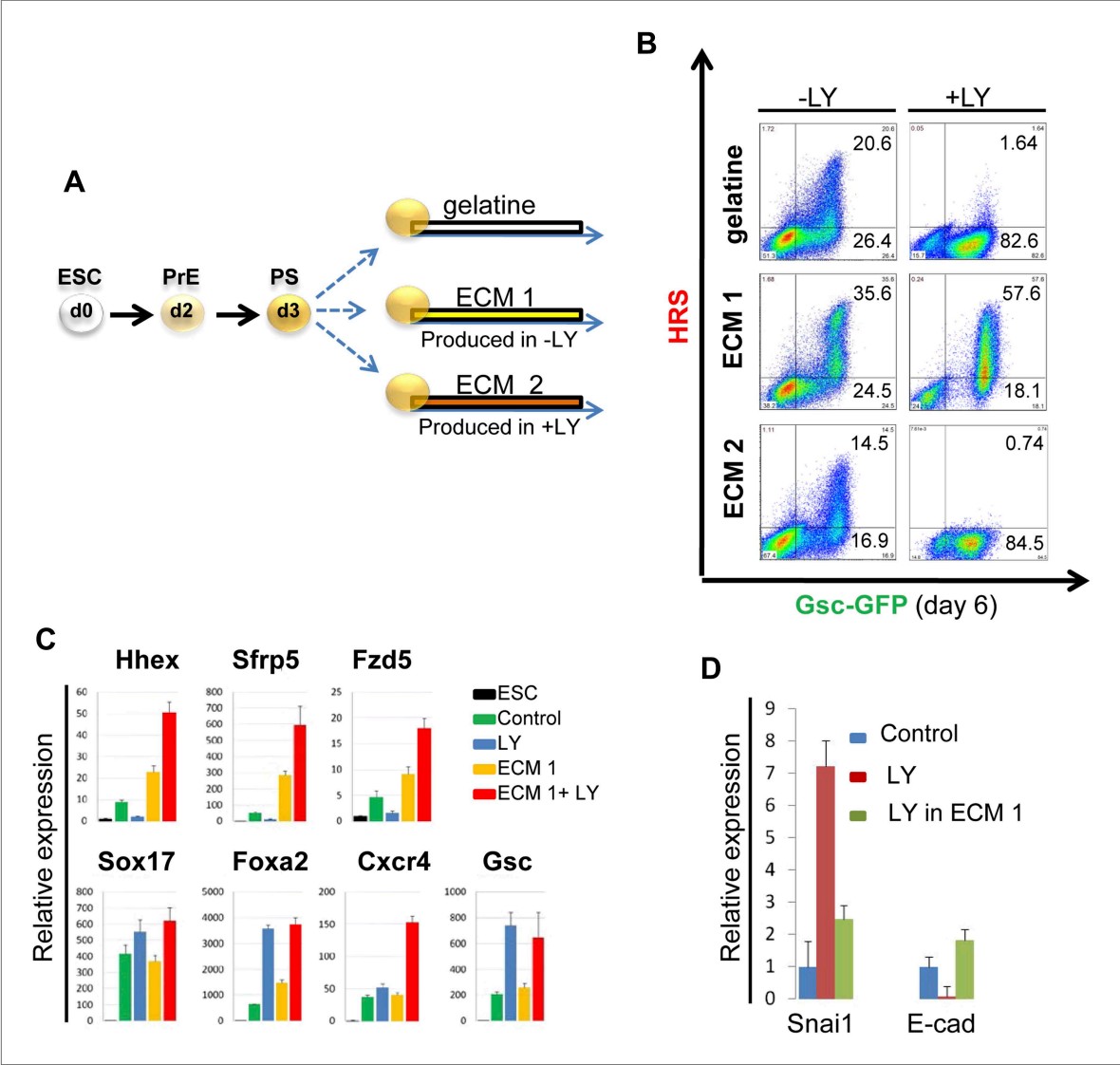

**Figure 6**. PI3K/Akt1 dependent-ECM generated during endoderm differentiation can support ADE specification. (**A**) Schematic showing the experimental strategy designed to test the role of ECM in supporting ADE specification. ECM1 and ECM2 were produced by differentiating cells in the absence or presence of LY respectively. Matrices were obtained by removing differentiated cells from day 6 cultures. (**B**) ADE generation assessed by flow cytometry on differentiating cells exposed to LY. ADE induction was rescued and enhanced when cells were re-plated onto ECM1. APS-like cells re-plated on either ECM2 or gelatine failed to counteract LY-induced phenotypes. (**C**) Q-RT-PCR analysis showing rescue of ADE markers in differentiating cells re-plated onto ECM1 and simultaneously treated with LY. Transcript levels were normalised to the *Tbp* value obtained for each sample. Normalised values are related to the level obtained for ESC. (**D**) Q-RT-PCR showing the regulation of EMT markers by ECM1. Transcript levels were normalised to the *Tbp* value obtained for each sample. Normalised values are related to the level obtained in control conditions.

The following figure supplements are available for figure 6:

**Figure supplement 1**. PI3K/Akt1 signalling modulates ECM activity to induce ADE specification.

or absence of LY. *Figure 6B* shows that ECM1, but not ECM2, could support ADE differentiation in the presence of LY. Moreover, ECM1 not only restored anterior induction, but the combination of LY and ECM1 enhanced anterior endoderm specification, such that these cultures were almost 60% ADE (*Figure 6B*). These data suggest that the capacity of LY to enhance Sox17⁺Foxa2⁺ expression was harnessed by ECM1, which was able to convert the Sox17⁺Foxa2⁺ naïve population into Hhex⁺ prospective foregut (*Figure 6C*). Cells plated on ECM1 in the presence of LY displayed epithelial morphology, enhanced E-cadherin and reduced Snai1 expression (*Figure 6D*, *Figure 6—figure*

supplement 1B). While treatment of differentiating cultures with LY resulted in a loss of expression of the tight junction and polarity markers, ZO-1 and aPKC, organized expression of these was restored within 12 hr of plating onto ECM1 (*Figure 6—figure supplement 1C*). Cells plated on ECM2 in the presence of LY did not generate ADE or form an organized epithelium (*Figure 6—figure supplement 1B*). To exclude the possibility that Akt1 activation was somehow stimulated by re-plating on these ECMs, despite the block to PI3K signalling, we assessed Akt1 activation after re-plating and observed no rescue of the LY-mediated block to Akt1 phosphorylation (*Figure 6—figure supplement 1D*). Thus, ECM1 has the capacity to redirect the LY-mediated enhancement of *Sox17/Foxa2* expression through an Akt1 independent pathway and convert the majority of the culture to properly patterned anterior endoderm. These data suggest that ECM1 together with LY could be used to enhance physiological ESC differentiation toward foregut (*Figure 6B*, middle panel).

## PI3K/Akt1 regulates fibronectin levels to specify endodermal identity

Gene expression profiling data (*Morrison et al., 2008*) suggested that components of the matrisome were up-regulated in ADE compared to ESC (*Figure 7—source data 1*), including a number of ECM proteases that proved to be PI3K-dependent (*Figure 7—figure supplement 1A*). To provide a better picture of PI3K/Akt1-dependent changes in the ECM that might be induced by post translational modification, we used mass spectrometry coupled to liquid chromatography (LC-MS) to define PI3K-dependent differences between ECM1 and ECM2. Details of ECM extraction and LC-MS methodologies can be found in 'Material and methods.' Specific proteins enriched in ECM1 include two heparan sulphates proteoglycans (HSPG), Perlecan and Col18a1; two HSPG regulators Sulf1 and Cyr61; the metalloprotease Adamts15, the anterior endoderm associated diffusible factors Nodal and Lefty1; and the endoderm/epithelial associated cytokeratines Krt8, Krt18 and Krt19 (*Figure 7A*, *Figure 7—source data 2,3*). While a number of these proteins are not canonical matrix proteins, they are generally associated with integrin signalling and tight junction formation. Interestingly, Fibronectin (Fn1), a promoter of EMT and migration, was highly enriched in ECM2 as was the PS associated diffusible factor Fgf8 (*Figure 7A*, *Figure 7—source data 2,3*). Analysis by q-RT-PCR confirmed that *Fn1* expression was transcriptionally regulated by PI3K/Akt1 signalling (*Figure 7B*), and that the induction of *Fn1* expression was a cell autonomous response to Akt1 activation (*Figure 7—figure supplement 1B*). Moreover, the addition of Fn1 to differentiating ESC cultures, either through re-plating of PS stage cultures onto Fn1-coated dishes (data not shown) or through direct addition to phase 2 differentiation (*Figure 7C*) abolished the induction of ADE specific transcription. As Fn1 is known to signal through Akt1 (*Khwaja et al., 1997*), and Akt1 is involved in ADE specification, we asked whether exogenous Fn1 resulted in Akt1 activation during normal endoderm differentiation. *Figure 7D* shows that Fn1 did not stimulate Akt1 phosphorylation, but it actually inhibited it (*Figure 7D*).

## PI3K/Akt1-dependent anterior specification and Fn1 expression in vivo

A number of ECM components identified by LC-MS were also screened by q-RT-PCR in different dissected regions of E7.5 embryos (*Figure 8—figure supplement 1A*). ECM1 components showed expression in the endoderm (*Figure 8—figure supplement 1B* and data not shown), as well as anterior enrichment (*Figure 8—figure supplement 1C* and data not shown). Importantly, we found that *Fn1* was both highly expressed in the endoderm (*Figure 8—figure supplement 1B*) and robustly enriched in the posterior region of these embryos (*Figure 8A,C–E*), consistent with a role in promoting naïve endoderm. To address whether endoderm regionalization and Fn1 expression were regulated in vivo by PI3K/Akt1, E6.5 embryos (initiation of gastrulation) were dissected and cultured ex-utero in the presence or absence of LY. Gene expression analysis in individual cultured embryos was assessed in response to inhibition of PI3K by LY. We observed enhanced expression of *Fn1* and *Snai1*, slight increases in *Sox17* and *Foxa2*, and a reciprocal reduction in *Cer1* and *Lefty1* (*Figure 8B*). Consistent with these observations, immunohistochemistry (IHC) showed that embryos cultured in the presence of LY exhibit enhanced Fn1 expression (*Figure 8C,D*) and a complete loss of Cer1+ ADE (*Figure 8—figure supplement 1D*).

Analysis of wild-type (WT) embryos at E7.5 indicated that Fn1 protein was non-uniformly distributed along the A-P axis, with relatively low expression in the prospective foregut region where Cer1 was expressed (*Figure 8A,C–E*). We therefore assessed the requirement of Fn1 during endoderm specification by examining the endoderm of embryos homozygous for a null mutation in the *Fn1* locus

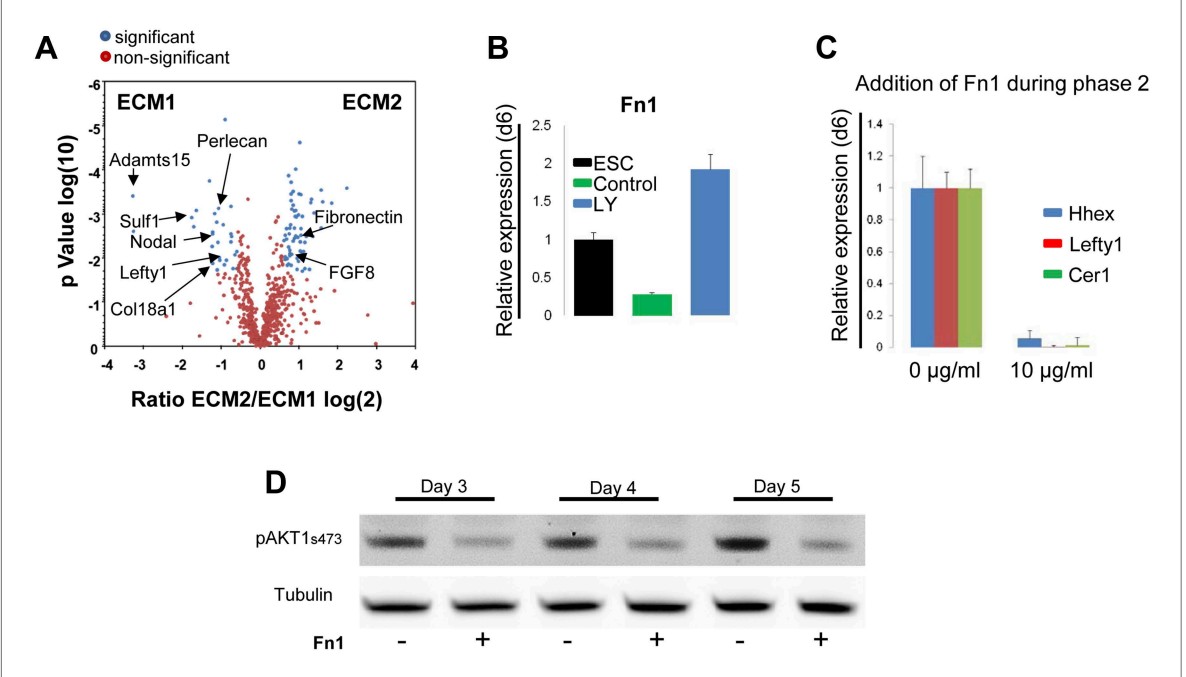

**Figure 7**. Composition of ECM determines cell fate choices within endoderm. (**A**) Different compositions of ECM1 and ECM2 as determined by mass spectrophotometry. Significantly over-represented peptides (blue dots) or evenly represented peptides (red dots) in the ECM1 (left side), and ECM2 (right side) are shown in the volcano plot. Relevant peptides are indicated. (**B**) Q-RT-PCR showing the regulation of Fn1 expression as a result of PI3K inhibition (d6). (**C**) Q-RT-PCR analysis showing inhibition of ADE gene expression when Fn1 was added to the culture media. Transcript levels were normalised to the *Tbp* value obtained for each sample. Normalised values are related to the level obtained in ESC (**B**) and in control conditions with no exogenous Fn1 (**C**). (**D**) Fn1 inhibits Akt1 activation. Western blots showing inhibition of Akt1 phosphorylation by Fn1 during ESC differentiation. Tubulin was used as loading control.

The following source data and figure supplements are available for figure 7:

**Source data 1**. ECM global gene analysis.

**Source data 2**. LC-MS analysis I.

**Source data 3**. LC-MS analysis II.

**Figure supplement 1**. Expression of ECM components during endoderm differentiation and in response to Akt1 activation.

(*Fn1⁻/⁻*) (*George et al., 1993*). In *Fn1⁻/⁻* embryos, the domain of Cer1 expression was expanded toward the posterior-proximal side of the embryo within the domain of Foxa2 expression (*Figure 8E*, *Figure 8—figure supplement 2A–C*). We also assessed the protein expression of the pan-endoderm marker Sox17 and found that like Fn1, Sox17 was expressed at lower levels in the Cer1 positive foregut region. In *Fn1* mutants, this region of Sox17[low] was expanded across the embryo alongside Cer1 (*Figure 8—figure supplement 3C*) and consistent with the increase in *Sox17* transcript that we have observed in LY-treated embryos. Moreover, while there are a number of integrins that bind to Fn1, integrin Itga5 is thought to mediate Fn1 functions during early embryogenesis and we found that *Itga5⁻/⁻* mutant embryos (*Yang et al., 1993*) also exhibited an expanded domain of Cer1 expression (*Figure 8—figure supplement 2D,E*). As the Fn1 and Itga5 phenotypes are known to be background dependent, we confirmed that we could observe expansion of the anterior endoderm domain in both *C57BL/6J* (*Figure 8E*, *Figure 8—figure supplement 2B,D*) and *129S4* mice strains (*Figure 8—figure supplement 2E*).

Consistent with the expansion of Cer1 in these mutants, further foregut morphogenesis was also defective and there was evidence of an expansion in the prospective foregut. In *Fn1* and *Itga5* null embryos: the foregut did not fold into a tube and remained as a flat field of Foxa2⁺ cells (*Figure 8F*,

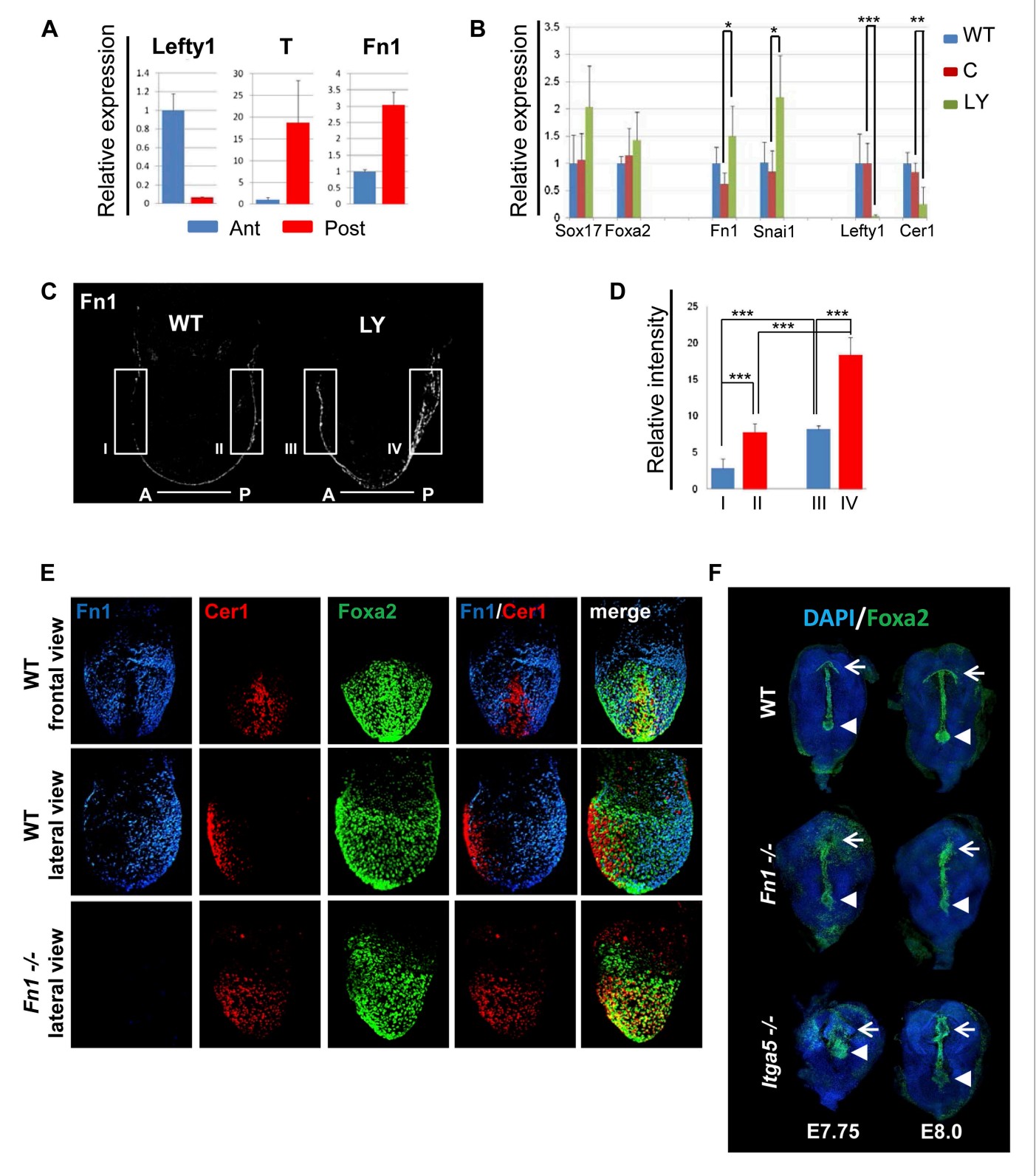

Figure 8. Anterior specification in vivo requires PI3K-dependent ECM. (A) Q-RT-PCR showing differential expression of *Fn1* in vivo. Individual E7.5 mouse embryos were bi-dissected into anterior and posterior halves, and *Lefty1* and *T* expression were used as a control for bi-dissection efficiency. Transcript levels were normalised to the *Tbp* value obtained for each sample. Normalised values are related to the level obtained in the anterior region. (B) *Fn1* is
*Figure 8. Continued on next page*

*Figure 8. Continued*

regulated by PI3K in vivo. Q-RT-PCR on single embryos dissected at E6.5 and cultured ex-vivo for 24 hr in the presence or absence of LY. Wild-type (WT) embryos were dissected at E7.5 and used as control. Embryos treated with LY exhibited a failure in the expression of anterior markers, and up-regulation of *Snai1* and *Fn1*, similar to that observed in vitro. Error bars represent the standard deviation between embryos (*>0.05; **>0.01; ***>0.001). Both in **A** and **B** the transcript levels were normalised to the *Tbp* value obtained for each sample. Normalised values are related to the level obtained in the anterior region (**A**) and in E7.5 wild-type embryos (**B**). (**C**) Immunohistochemistry (IHC) on sectioned embryos showing higher levels Fn1 on the posterior side of E7.5 wild-type embryos (left) and increased expression of Fn1 in embryos dissected at E6.5 and cultured ex-vivo during 24 hr in the presence of LY (right). (**D**) Images analysis on the areas highlighted in (**C**) indicating differential expression of Fn1 along the A–P axis in WT embryos (I, II), and increased expression of Fn1 in embryos cultured ex vivo in the presence of LY (III, IV) (n = 4, ***>0.001). (**E**) IHC on E7.5 embryos showing differential expression of Fn1 along the anterior–posterior axis and anterior expression of Cer1. In *Fn1*−/− embryos the Cer1 domain is significantly expanded. Embryos are shown in frontal and in lateral view (anterior to the left). (**F**) IHC on WT and mutant embryos (dorsal view) showed an expanded Foxa2 expression domain at E7.75, and defects in gut folding and in the formation of the anterior intestinal portal (AIP) at E8.0. Arrowhead points to the node and arrows the foregut.

The following figure supplements are available for figure 8:

**Figure supplement 1**. Composition of ECM in the gastrulation stage mouse embryo.

**Figure supplement 2**. Reduction in Fn1 activity resulted in expansion of the anterior endodermal domain in mouse gastrula.

**Figure supplement 3**. Fibronectin mutants exhibit defects in foregut and naïve endoderm.

*Figure 8—figure supplement 3B*). This domain of Foxa2+ cells remained enlarged compared to control embryos and this was particularly apparent early in foregut morphogenesis (E7.75) (*Figure 8F*, *Figure 8—figure supplement 3A,B*).

## Discussion

We have defined a novel ECM-dependent mechanism that can explain the diverse developmental outcomes induced by a single signalling pathway. Thus, a requirement for FGF signalling in endoderm induction is based on a non-canonical activity downstream of PI3K/Akt1 signalling. PI3K/Akt1 acts during positional specification in the endoderm, by regulating the deposition of regionalized ECM with patterning activity (*Figure 9*). One essential component of this activity is Fn1. This observation led to the identification of PI3K/Akt1-dependent graded Fn1 expression within the developing embryo with the capacity to differentially regulate the timing of endoderm differentiation and establish positional identity.

Mesendoderm induction is associated with specific levels of Nodal, Wnt and FGF signalling, but is also accompanied by morphogenetic rearrangements that result in changes in a cell's immediate environment. These cellular movements, the ECM that cells move on and the signals they are exposed to during their migration all contribute to the patterning of the embryonic endoderm (*Arnold and Robertson, 2009*). The non-canonical ECM-dependent pathway downstream of PI3K described here provides an explanation for the diversity of cellular responses that can be achieved through the activation of pAkt1. PI3K/Akt1 signalling has been extensively associated with the induction of EMT (*Larue and Bellacosa, 2005*) rather than its suppression. However, in the context described here, Akt1 modifies the ECM to drive cell–cell contact and either induce or preserve epithelial identity. Thus, high levels of PI3K activity ensure that Fn1 levels remain low and that EMT is suppressed. The involvement of an ECM intermediate also provides context for an endoderm specific role for FGF signalling. FGF receptor activation is essential for ESC differentiation towards multiple lineages and leads to Grb2 and SHP2 recruitment, activating both PI3K and MAPK signalling pathways (*Kouhara et al., 1997*; *Xu et al., 1998*; *Villegas et al., 2010*). These pathways have been associated with mesoderm induction, EMT and cell migration during gastrulation (*Casey et al., 1998*; *Rodaway et al., 1999*; *Ciruna and Rossant, 2001*; *Mizoguchi et al., 2006*; *Poulain et al., 2006*). In endoderm, where FGF signalling is required, but does not induce EMT (*Morrison et al., 2008*; *Hansson et al., 2009*), the signalling is contextualized and rendered distinct as a result of the activity of this pathway in remodelling the ECM. PI3K activity in the visceral endoderm of embryoid body differentiation (*Li et al., 2001*) has also been associated with the production of ECM required to support the epithelialization of the epiblast (*Jeanes et al., 2009*).

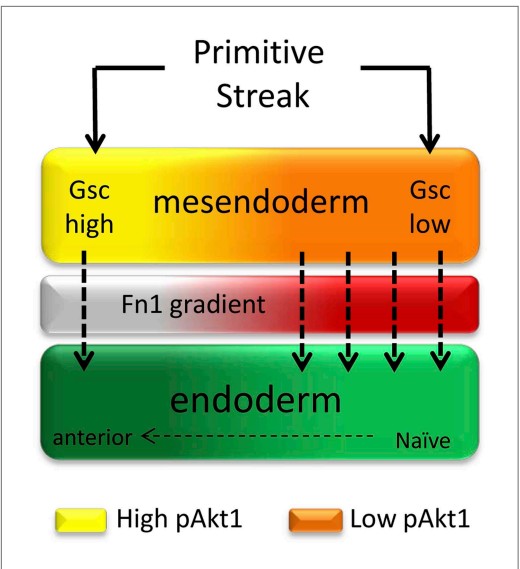

**Figure 9**. A role for ECM in positional specification in the endoderm. Schematic illustration of the proposed role played by PI3K/Akt1 signalling in positional identity within the endoderm. Cells in PS region are in the process of generating mesoderm and endoderm (mesendoderm). At the anterior end of the PS, high levels of pAkt1 regulate proper epithelialization and anterior endoderm specification through the production of a unique ECM low in Fn1. Low levels of PI3K/Akt1 led to the production of an Fn1-enriched ECM that favours naïve endoderm generation.

While the activity of PI3K/Akt1 is generally associated with induction of EMT, activation of this pathway has also been associated with a block to EMT in human ESCs (*Singh et al., 2012*). In this context, Akt1 activation supports human ESC self-renewal by inhibiting Raf/Mek/Erk and canonical Wnt signalling, which in turn may also promote EMT, although it is not clear whether this route takes place in vivo during early endoderm differentiation or how comparable the mouse and human ESC differentiation models are. Accordingly, the transient reduction in pAkt1 we observed at the beginning of differentiation may be coupled to early activation of the ERK and Wnt pathways as cells become prepared for mesoderm and endoderm induction. However, prior to the appearance of endoderm gene expression, differentiating ESCs contained high levels of pAkt1 and this appeared essential for the deposition of the correct, Fn1$^{low}$ regionalized ECM. That cells rapidly down-regulate Snai1 upon replating suggests that the ECM-dependent pathway described here allows rapid modifications to differentiation as cells become exposed to a new environment.

The basement membrane produced during endoderm differentiation is complex and proteomic analysis supports the association of the ECM with functional cell membrane components in a matrisome (*Hynes and Naba, 2011*). How do these elements impact on the key signalling events regulating lineage specification? Several elements of this matrisome are directly involved in Fn1 signalling and exogenous Fn1 can alter ECM activity. Genetic studies indicate that Fn1-integrin interactions are essential for axial extension of the mesendoderm (*Yang et al., 1999*; *Davidson et al., 2006*; *Marsden and DeSimone, 2003*). However, Fn1 is not required for gastrulation and specific endodermal patterning defects have not been previously reported. The *Fn1* and *Itga5* null mutants are both embryonic lethal and die by E10.5 due to cardiovascular problems, posterior truncations, and kinking of the neural tube among others defects (*George et al., 1993*; *Yang et al., 1993*). We observed higher levels of Fn1 in the posterior region of gastrulation stage embryos and the region where the nascent foregut moves into was uniquely low in Fn1. In Fn1$^{-/-}$ embryos, the region of Cer1 positive ADE is expanded, suggesting that the deposition of Fn1 may help to restrict the prospective foregut domain as it migrates up from the distal end of the embryo. Consistent with this observation, the foregut did not fold into a tube and remained as a flat field of expanded Foxa2$^+$ cells in Fn1$^{-/-}$ embryos, indicating that the regulation of Fn1 deposition helps delineate and form the prospective foregut. We also observed a similar inverse correlation between the expression of Sox17 and Cer1, indicating that Sox17 is expressed at higher levels in Fn1 rich regions of the endoderm. This is consistent with our in vivo and in vitro data, which show significantly higher levels of Sox17 expression in naïve endoderm that was generated in response to inhibition of PI3K.

In addition to Fn1, we identified additional potential determinants that could interact with Fn1 in modulating ECM activity. Two out of the three main Heparan Sulphate-bearing species, Perlecan and Col18a1 (*Halfter et al., 1998*) were expressed at high levels in ECM1. *Knock down* of these two HSPG's have been shown to impair endoderm differentiation in vitro (*Higuchi et al., 2010*). Potentially, this HSPG-bearing ECM could be acting in conjunction with low levels of Fn1 to modulate growth factor bioavailability, and operate as a platform for cytokine signalling. As the cytokines we added exogenously to the cultures were not found in either ECMs, we think it is unlikely that this explains the

full activity of these ECMs in vitro. However, we did find that ECM1 contained Nodal and Lefty1, whereas ECM2 contained Fgf8 (*Figure 7A*, *Figure 7—source data 2,3*). It has been suggested that specific modifications in the ECM are required for Nodal transport within the embryo (*Oki et al., 2007*; *Marjoram and Wright, 2011*). Moreover, the inclusion of Nodal, Lefty1 and Fgf8 within these matrices fits with the locus of their extracellular activity in vivo. Our observation that these cytokines appear specifically embedded in the ECM suggests that the mechanism by which these proteins function in vivo involves association with ECM proteins (*Hynes, 2009*).

Despite widespread attempts to use ESC differentiation towards endodermal derivatives, the production of fully functional differentiated cell types has not been particularly successful. Part of the explanation for this lack of success may lie in the inability to pattern the correct intermediates. The observation that inhibition of PI3K leads to enhanced Sox17 and Foxa2 expression (*McLean et al., 2007*) has been used in ESC differentiation protocols to generate endoderm derivatives. Our data suggest that these cells are less efficient in further differentiation to derivatives of the ventral foregut such as pancreatic or hepatic progenitors. Based on both molecular and functional analysis of these endoderm cells they appear to be naïve endoderm, similar to the mesenchymal cell type that emerges from the primitive streak and intercalates into the visceral endoderm layer (*Kwon et al., 2008*; *Burtscher and Lickert, 2009*) to form the hindgut. These $Sox17^{high}$ mesenchyme-like cells that arise from the primitive streak region of the embryo exist only transiently in vivo, and while we show that prolonged and enhanced induction of this state in vitro may be reversed by reintroducing cells back into normal differentiation, the direct onward differentiation of these cells may have deleterious effects on their capacity to form functional cell types in vitro. Interestingly, while these cells become epithelialized as they intercalate in the visceral endoderm, their levels of Sox17 remain high throughout gastrulation, suggesting that a naïve endoderm state could extend beyond the initial EMT event.

As ECM1 is able to pattern differentiating cells to anterior fate, it suggests that it may be possible to use it in combination with the LY-dependent enhancement of naïve endoderm to efficiently generate high levels of foregut in vitro. Our finding that ECM is an essential downstream signalling response required for generation of positional identity in vitro indicates that the presence of appropriate ECM may be as important as the dose and identity of the cytokines used in directing context dependent signalling responses during ESC differentiation.

## Materials and methods

### Cell culture

Mouse ESC cultures and ADE differentiation in adherent monolayer were performed according to *Livigni et al. (2009)* except that in some cases SFO3 media was replaced by ADEM (see below). Re-plating of APS-like cells was done by differentiating the HRS/Gsc-GFP ESC until Gsc-GFP expression became apparent (to day 3–3.5). The media was collected, centrifuged at 3000 rpm for 8 min and mixed with freshly prepared media in ratio 1:1 (new:old). The APS-like cells were detached with Accutase (Sigma), 5 min at 37°C, washed with the collected media, resuspended in 1:1 media, and re-plated onto different substrates.

For the induction of pancreatic progenitor cells, ADE and LY-cells in N2B27 media supplemented with 10 ng/ml FGF4, 2 mM all-trans retinoic acid, and 0.25 mM KAAD cyclopamine (Toronto Research Chemicals). Hepatocyte differentiation was as described in *Gouon-Evans et al. (2006)*, except N2B27 that was used as the base media at all stages and Activin A was not included, whereas BMP4, FGF2, and VEGF were included.

ADEM (Anterior definitive endoderm media) preparation: 50% DMEM/F12 no Glutamine (Invitrogen, UK), 50% RPMI1640 no Glutamine (Invitrogen, UK), 2 mM L-Glutamine (Invitrogen, UK), 10 mM HEPES (Invitrogen), 0.5 µg/ml Cholesterol (Sigma-Aldrich, Denmark), 80 µM Ethanolamine (Sigma-Aldrich, Denmark), 10 µg/ml Cytidine (Sigma-Aldrich, Denmark). Filter sterilize. 2 µg/ml Apo-transferrin (Sigma-Aldrich, Denmark), 2 ng/ml Na Selenite (Sigma-Aldrich, Denmark). Just before use add: 0.1 µM 2-ME (Sigma-Aldrich, Denmark), 0.1% BSA (Invitrogen, UK), 0.5 µg/ml human recombinant insulin (Sigma-Aldrich, Denmark). Media must be kept at 4°C and used within 1 month.

### ECM isolation and collection

Differentiated ADE cells were detached with 2 mM EDTA, 37°C, 8–12 min. The remaining ECM was washed with PBS and either used immediately or preserved in 70% EtOH, −20°C.

## Inhibitor manipulation

Doses and timing of inhibitor usage are described in *Supplementary file 1*.

To determine the window of PI3K activity during phase 2 of ADE differentiation LY (LY294002; Promega, UK) was added for periods of 24 hr (d3–4, d4–5; d5–6, d6–7), 48 hr (d3–5, d4–6, d5–7), 72 hr (d3–6, d4–7), or 96 hr (d3–7). ADE generation was assessed by flow cytometry based on HRS/Gsc-GFP expression. LY was used at a concentration of 10 µM during d3–4 and at 20 µM from d4. 20–50 µM LY was used for experiments assessing apoptosis from d3-d6.

For co-culture experiments, HRS and Akt1-GFP-HRS ESC were mixed in a 1:1 ratio. 0.5 µM Tam was added 15 min before LY addition at day 3.

For ex-vivo gastrula cultures, embryos were dissected at E6.5 and cultured in 25, 50, or 75 µM LY for 24 hr. 25 µM and showed no morphological or differences in housekeeping gene transcription compared to untreated embryos. While 50 µM LY showed a significant difference compared to untreated embryos, 50 and 75 µM LY gave similar results (Data not shown), and 50 µM LY was therefore used for further experiments.

Apoptosis inhibitor (Z-VAD-FMK; Santa Cruz, USA; 10 µM) was added at d3–6. The effects of LY or Z-VAD-FMK addition were analysed by immunostaining with the apoptotic marker Annexin-V Alexa647 (Invitrogen, UK) followed by flow cytometry analysis (as described below).

## Flow cytometry and cell sorting

Cells were trypsinized and stained with Topro3-iodide or DAPI (Invitrogen, UK) to exclude dead cells from the analysis. FITC-conjugated rat anti-CD184 (Cxcr4) was purchased from BD PharMingen (UK), cells were analysed using a Becton Dickinson FACS Aria II or III cell sorter and a Becton Dickinson LSR Fortessa.

## Gene expression analysis

Total RNA was prepared from a minimum of $1 \times 10^4$ cells using Trizol reagent (Invitrogen) and 1 µg of RNA was used as a template for cDNA synthesis using Superscript III (Invitrogen, UK). Real-time RT-PCR was performed using a LightCycler 480 (Roche) and LightCycler 480 SYBR Green 1 Master or UPL Assay (Roche). Primers and PCR conditions are listed in *Supplementary file 3*.

## Immunostaining

Monolayer differentiated cells were washed in PBS; fixed in 4% paraformaldehyde (PFA) for 10 min; 1M Glycine, pH 7,4 was used for 10 min to block residual PFA, and cells were permeabilized by washing in PBS supplemented with 0.1% Triton X (PBST). The fixed cells were blocked in 1% Bovine serum albumin (Sigma-Aldrich, Denmark) and 3% appropriate serum in PBST for 1 hr at room temperature (RT).

For whole mount IHC, embryos were dissected in ice-cold M2 media and transferred to cold 4% PFA, overnight (O/N). The embryos were dehydrated in methanol series (25, 50, 75, 100%, in PBS), bleached with 5% $H_2O_2$ for 1 hr, rehydrated, blocked with 3% of the appropriate serum in PBST, incubated with primary antibody (in PBST at 4°C, O/N), washed in PBST 3 × 15 min, 5 × 1 hr, incubated with secondary antibody for 2 hr, RT, washed in PBST 3 × 15 min, 5 × 1 hr. For confocal imaging embryos were placed in microscope slides mounted with Vectastain/DAPI (Vector Labs) and gently covered with glass cover slips with vaseline drops in the corners to avoid embryo squashing.

For cryo-sectioned embryos and IHC, whole-mount embryos were dissected and fixed as above and allowed to sink in 15% sucrose in PBS, 2 hr at 4°C, then incubated in 15% sucrose/7% gelatine/PBS at 37°C, assembled in an aluminium mould, frozen in liquid nitrogen, and cut into 10 µm sections on a Cryostat (Leica). Both transverse and sagittal sections were the same thickness and were counterstained with DAPI. Sections were collected on Poly-Lysine microscope slides (VWR International), air-dried for 30 min to 1 hr, and stored at −20°C until use. Immunocytochemistry was performed essentially as described above for cells. A list of antibodies and conditions used is provided in *Supplementary file 2*.

## Western blots

Cells were lysed in lysis buffer (1% Triton X-100, 150 mM NaCl, 10 mM Tris_HCl, at pH 7.4, 1 mM EDTA, 1 mM EGTA, containing 2 mM NaF, 1 mM sodium orthovanadate, 10 µg/ml leupeptin, 10 µg/ml pepstatin,10 µg/ml aprotinin, and 1 mM Pefabloc). Equal amounts of protein lysates were loaded and

separated by SDS-PAGE and followed by western blotting. A list of antibodies used is provided in *Supplementary file 2*.

## Imaging

Whole-mount embryos (E7.5) were photographed on an AZ100 Multizoom microscope (Nikon) with DIC optics and standard epifluorescence using an EXI-BLU-R-F-M-14 camera (QImaging, Canada) and Volocity software (Improvision). For higher resolution, DMIRE2 inverted confocal (Leica) microscope was used. For confocal imaging, a series of 5 µm optical sections were taken and deconvolved with Volocity or equivalent software according to the instructions of the manufacturer.

## Embryo dissection and culturing

For A–P expression analysis, E7.5 embryos were dissected out of decidua in ice-cold M2 media (Sigma), and bisected into anterior and posterior halves with glass needles after removal of the extraembryonic tissues. Individual halves were prepared for RNA isolation with Trizol reagent (Invitrogen). For endoderm dissection, E7.5 embryos were dissected as stated before and the embryonic fragments were incubated for 10–15 min at 4°C in a solution of 0.5% trypsin and 2.5% pancreatin in PBS. The embryos were drawn into a hand-pulled glass pipette with a diameter slightly smaller than the fragment to peel away the endoderm and separate it from the epiblast. Both fragments were recovered and prepared for RNA isolation. For embryo culture, E6.5 mouse embryos were dissected as stated above, but leaving the ectoplacental cone intact and removing only Reichart's membrane. The embryos were cultured in 1:1 rat serum/GMEM media (1 ml per embryo) supplemented with 200 mM L-glutamine/100 mM sodium pyruvate (Millipore, UK), 25 U/Ml Penicillin/25 µg/Ml Streptomycin, treated or not with 50 µM LY, in rotating wheel incubator at 37°C, 5% $CO_2$. After 24 hr, the embryos were collected and fixed with 4% PFA for IHC or prepared for RNA isolation. The embryos from the same litter (WT) were dissected at E7.5 and used as wild-type controls. T-test was used to analyse the relationship between groups. WT, n = 12; C, n = 10; LY, n = 9; p values: *$p < 0.05$, **$p < 0.01$, or ***$p < 0.001$.

## LC-MS analysis

For ECM analysis samples were obtained and prepared according to an adaptation from *Turoverova et al. (2009)*. Briefly, cells were removed by 2 mM EDTA (500 µl per 35 mm well), 37°C, 8–12 min, without shaking, and the cell detachment was controlled by eye. The solution was gently aspirated and residual cells and debris removed by 1×PBS wash/aspiration. Then the wells were covered with 800 ml 5% acetic acid, 4°C, ON. The wells were scrapped, the solution collected, and the wells covered with extraction buffer: 125 mM tris-HCL, pH 6.8, 1% SDS, 20 mM DDT, 10% glycerol, 0.05 mM PSMF, protease inhibitor cocktail (Santa Cruz Biotech, USA), and placed at 37°C, on a shaking platform incubator, 2 × 1 hr. Proteins were scrapped, collected and combined with the acetic acid fraction, and precipitated using methanol:chloroform. Briefly, equal volumes of protein extract and methanol were mixed together with one quarter of volume of chloroform. The mixture was centrifuged at 9000×*g* for 1 min and the upper phase was removed. At least 3 volumes of methanol was added to the lower phase and interphase with precipitated protein, mixed and centrifuged at 9000×*g* for 2 min. The pellets were allowed to air dry before solubilisation. Protein concentration was measured using the Amido Black Protein Assay (*Dieckmann-Schuppert and Schnittler, 1997*).

Proteins were dissolved in 200 µl of HEPES buffer (200 mM HEPES, pH 8.0 containing 0.1% of RapiGest SF) (Waters Corp., UK). Protein sample was reduced in 5 mM THP for 30 min at 37°C, and cysteines were alkylated in 10 mM iodoacetamide at RT in the dark for 30 min. Trypsin (modified, sequencing grade, Roche) was added to the sample at a ratio of 1:50 enzyme/protein, and allowed to digest overnight at 37°C. After tryptic digestion, formic acid was added to 2% final concentration to stop the reaction, and samples were then incubated at 37°C for an additional 4 hr. The samples were centrifuged for 30 min at 10,000×*g* to remove insoluble material and filtered with a 0.2 µm cartridge (Varian, UK). Filtered peptides were dried by a Speed Vac, and stored at −20°C.

Capillary-HPLC-MSMS analysis was performed on an on-line micro-pump (1200 binary HPLC system, Agilent, UK) coupled to a hybrid LTQ-Orbitrap XL instrument (Thermo-Fisher, UK). The LTQ was controlled through Xcalibur 2.0.7 and LTQ Orbitrap XL MS2.4SPI. HPLC-MS methods have been described previously (*Le Bihan et al., 2010*). Samples were reconstituted in 10 µl loading buffer before injection, and analysed on a 2 hr gradient for data dependant analysis.

As a control, pure gelatine was treated and analysed using the same procedure. *Figure 7—source data 2* also shows the results for gelatine MS-LC analysis.

## Data analysis

LC-MS Label-free quantification was performed using Progenesis 2.6 (Nonlinear Dynamics, UK). The number of Features was reduced to only MSMS peaks with a charge of 2, 3, or 4+ and the five most intense MSMS spectra per 'Feature' were kept. The generated MGF files were searched using MASCOT Versions 2.3 (Matrix Science Ltd, UK) against a mouse plus contaminant proteins IPI database with 55413 sequences downloaded from www.ebi.ac.uk (version v3.42). Variable methionine oxidation, STY phosphorylation, protein N-terminal acetylation and fixed cysteine carbamidomethylation were used in all searches. Precursor mass tolerance was set to 7 ppm and MSMS tolerance to 0.4 amu. The significance threshold (p) was set below 0.05 (MudPIT scoring in Mascot). The list of proteins and quantitation are reported in Supplemental Information. Q values for false discovery rate control based on p value generated by Progenesis software were calculated from a R package (*Storey and Tibshirani, 2003*). A similar approach to *Hartmann et al. (2009)* in terms of threshold choice was used in this study. Hierarchical clustering was performed using Genesis (*Sturn et al., 2002*) with an average linkage clustering.

## Acknowledgements

We wish to thank Takumi Era and Shinichi Nishikawa for the kind gift of the Goosecoid-GFP targeting construct and technical advice, and Tohru Kimura and Tom Burdon for the Myr-Akt1-mER vector. We thank Korinna Henseleit and Afifah Rahman for generating the HRS/Gsc-GFP cell line, its construction will be described elsewhere. We thank Tristan Rodriquez, Henrik Semb, Sally Lowell, Diana Escalante-Alcalde, Anne Grapin-Botton and the entire Brickman lab for critical comments and discussions on this manuscript. JMB was supported by an MRC Senior Non-Clinical Fellowship.

## Additional information

### Funding

| Funder | Grant reference number | Author |
|---|---|---|
| Medical Research Council | G0701428 | S Nahuel Villegas, Joshua M Brickman |
| Novo Nordisk Foundation, Section for Basic Stem Cell Biology | | Joshua M Brickman, Michaela Rothová, S Nahuel Villegas |
| Biotechnology and Biological Sciences Research Council and the Engineering and Physical Sciences Research Council | BB/D019621/1 | Thierry Le Bihan, Martin E Barrios-Llerena |
| National Heart, Lung, and Blood Institute | 5RO1HL103920 | Sophie Astrof |
| American Heart Association Innovative Research Grant | 12IRG9130012 | Sophie Astrof |
| National Institutes of Health | RO1-DK084391 | Anna-Katerina Hadjantonakis, Maria Pulina |

The funders had no role in study design, data collection and interpretation, or the decision to submit the work for publication.

### Author contributions

SNV, Conception and design, Acquisition of data, Analysis and interpretation of data, Drafting or revising the article; MR, SA, Acquisition of data, Analysis and interpretation of data, Drafting or revising the article; MEB-L, Acquisition of data, Analysis and interpretation of data; MP, Acquisition of data; A-KH, Analysis and interpretation of data, Drafting or revising the article; TLB, Conception and design, Analysis and interpretation of data; JMB, Conception and design, Analysis and interpretation of data, Drafting or revising the article

## Ethics

Animal experimentation: All animal work was conducted in accordance with UK and European legislation and, in particular, according to the regulations described in the Animals (Scientific Procedures) Act of 1986 (UK). All work in this manuscript was authorized by and carried out under Project License 60/3715 issued by the UK Home Office. Genetic modification for the generation of mouse ESC lines was approved by the ethics committees of the University of Edinburgh and the University of Copenhagen.

## Additional files

### Supplementary files

• Supplementary file 1. Inhibitor doses and cell survival.

• Supplementary file 2. List of primers and conditions used.

• Supplementary file 3. List of antibodies and conditions used.

### Major dataset

The following previously published dataset was used:

| Author(s) | Year | Dataset title | Dataset ID and/or URL | Database, license, and accessibility information |
|---|---|---|---|---|
| Morrison GM, Oikonomopoulou I, Portero Migueles R, Soneji S, Livigni A, Enver T, Brickman JM | 2008 | Transcription profiling of Hex+Cxcr4+ differentiating mouse embryonic stem cells to study a role for FGF signalling in the generation of Anterior Definitive Endoderm | E-TABM-515; http://www.ebi.ac.uk/arrayexpress/experiments/E-TABM-515/ | Publicly available at Array Express (http://www.ebi.ac.uk/arrayexpress/). |

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
