## [Decision Letter]

Thank you for sending your work entitled “PI3K/Akt1 signalling specifies foregut precursors by generating regionalized extra-cellular matrix” for consideration at *eLife*. Your article has been favorably evaluated by 3 reviewers, one of whom is Senior editor Janet Rossant.

The Senior editor and the other reviewers discussed their comments before we reached this decision, and the Senior editor has assembled the following comments to help you prepare a revised submission.

The reviewers all agreed that this paper adds significant new information to our understanding of early endoderm patterning in ES cells and the embryo. They have some areas of concern that would need to be addressed before the paper can be accepted for publication, as listed below. The two most important relate to the alternate fates adopted after Ly treatment and a further elucidation of the relationship between the phenotype of Fn^-/-^ embryos and the effects described here on endoderm patterning.

1) It is stated that treatment with LY produces cells that are either more posterior endoderm-like, or more like undetermined early endoderm. Which is actually the case? How can this be distinguished? It is an important point, because it could be that LY treatment just blocks further differentiation of all endoderm, rather than specifically blocking anterior endoderm formation. A more careful analysis of the alternative fates adopted would be useful.

2) It is also stated that high levels of LY caused cell death, which is perhaps not unexpected given the fact that Akt signaling is often associated with cell survival. How do the authors distinguish between effects on cell differentiation and cell survival in their system, in both ES cells and embryos? Is there significant cell death in ES cells and embryos treated with the dose of LY that seems to block ADE formation?

3) The data presented on the Fn^-/-^ embryos is intriguing but limited. They show expansion of the Cer1-expressing domain, consistent with their hypothesis that low levels of Fn are associated with ADE formation. However, they do not follow this further to determine whether there is any effect on actual foregut formation.

4) The phenotypes of Fn^-/-^ embryos have been reported to be very variable dependent on genetic background, and so it is important to understand the overall phenotype seen in their particular mutant strain.

5) In the discussion, they state that there is no gastrulation phenotype in Fn mutants, and yet they are actually reporting such a phenotype with the Cer1 expansion. Clearly a further analysis of the effect of the Fn mutation on gastrulation, mesoderm and endoderm formation is called for.

6) Despite their emphasis in the results on FN being the key determinant of the patterning of the endoderm, in the discussion they seem less sure and talk about the complexities of the ECM differences. This part needs some clarification.

---

## [Author Response]

*1) It is stated that treatment with LY produces cells that are either more posterior endoderm-like, or more like undetermined early endoderm. Which is actually the case? How can this be distinguished? It is an important point, because it could be that LY treatment just blocks further differentiation of all endoderm, rather than specifically blocking anterior endoderm formation. A more careful analysis of the alternative fates adopted would be useful*.

We thank the reviewers for asking us to clarify this and we have added additional data to address this point.

• We have added additional molecular analysis on differentiating cultures to show that the markers expressed by LY-treated cells represent early naïve endoderm similar to that induced in the primitive streak of the embryo (Figure 3—figure supplement 1). Based on this, and our previous analysis (Figure 2), these cells expressed neither anterior (*Cer1, Hex, Lefty, Sfrp5 and Fzd5*) posterior (*Cdx2*), or visceral (*HNF4a, Dab2*) endoderm markers. They expressed high levels of the endodermal markers Sox17 and FoxA2, alongside primitive streak markers *MixL1* and *Cdh2*. These cells also express high levels of the mesenchymal marker *Snai1*, but did not express the mesoderm markers Mesp1, Mesp2, Isl1, Gata1, Meox1 and Meox2. While LY treatment led to an increase in primitive streak markers, the primitive streak cells in these cultures (81% of cells were Gsc positive primitive streak – Figure 3—figure supplement 1) were almost all endoderm (91% Sox17 positive).

• We have also added additional functional data to assess whether these naïve cells could still progress in differentiation after the PI3K block was removed. In the new Figure 3 we show that these cells can return to normal ADE differentiation and with an equivalent amount of extra time, will “catch up,” with normal cultures.

• We also tested whether the LY treated “naïve” endoderm could efficiently differentiate towards pancreatic and hepatic endoderm (new Figure 3, and Figure 3—figure supplement 1). In both cases the Sox17 positive naïve endoderm was less efficient in downstream differentiation. However, when these cells were returned to normal ADE culture to enable them to “catch up” prior to further differentiation, these cells could efficiently differentiate towards HNF4a/AFP positive hepatic endoderm (new Figure 3).

• We also subjected the cultures to intestinal differentiation (38), but we were not able to generate intestine spheres from either ADE or LY-treated cells. This may indicate that specific posterior pre-patterning of naïve endoderm is required.

• Taken together, this new set of experiments suggest that the cells produced by PI3K/Akt1 inhibition are non-regionalized endoderm (naïve) that retain the ability to give rise to specific endodermal domains. All this new analysis is described and discussed in the new manuscript.

*2) It is also stated that high levels of LY caused cell death, which is perhaps not unexpected given the fact that Akt signaling is often associated with cell survival. How do the authors distinguish between effects on cell differentiation and cell survival in their system, in both ES cells and embryos? Is there significant cell death in ES cells and embryos treated with the dose of LY that seems to block ADE formation*?

Again, we thank the reviewers for asking us to clarify this point as it prompted us to generate additional new data that adds to the paper. These data are discussed in the Results section.

• We agree that cell death caused by high levels of LY is expected due to the role of PI3K/Akt1 signaling in cell survival. For this reason we only used low and medium doses of LY in this study (low = 10 µM, medium = 20 µM).

• We have now included data assessing cell death analysis in differentiating cultures. We used Annexin V staining of Gsc positive and negative populations to determine whether the LY doses used in our experiments (20 µM LY) had a specific effect on cell death. High dose (50 µM) was used as control (new Figure 2—figure supplement 1). Interestingly, we found enhanced cell death in the Gsc negative population of undifferentiated cells at day 3, while enhanced cell death in differentiating Gsc positive populations was not detected. From day 4 onwards neither population was significantly affected. To test if the enhanced cell death observed at day 3 on Gsc negative cells had a role in the LY-induced block to anterior differentiation, we suppressed cell death by the addition of Z-VAD-FMK (10 µM), a specific Caspase inhibitor. However, Z-VAD-FMK failed to rescue the block to ADE specification generated by LY. Based on this new data we can conclude that blocking PI3K signaling is not mediating differentiation via selective apoptosis.

• Although cell death was not specifically assessed in LY-treated embryos, we believe our in vitro data is sufficient to demonstrate that LY was not leading to the death of ADE progenitors. However, DAPI nuclear counterstaining also allowed us to evaluate the level of apoptotic cell bodies present in the embryos. These apoptotic cell bodies are easily identifiable by microscopy. We observed no differences in the number of apoptotic cell bodies in LY-treated and not treated cultured embryos.

*3) The data presented on the Fn*^*-/-*^
*embryos is intriguing but limited. They show expansion of the Cer1-expressing domain, consistent with their hypothesis that low levels of Fn are associated with ADE formation. However, they do not follow this further to determine whether there is any effect on actual foregut formation*.

• While no foregut phenotype has been reported previously for *Fn1* or *Itga5* mutants, we want to thank the reviewers for pushing us to examine the phenotype more extensively. We have added additional data based on this comment, describing a later phenotype in the foregut of *Fn1* and *Itga5* mutants (new Figure 8, Figure 8—figure supplement 3). In embryos mutant for either *Fn1* or *Itga5* the foregut does not fold into a tube and remains as a flat field of Foxa2^+^ cells (Figure 8, Figure 8—figure supplement 3).

• We have now added Figure 8 (Figure 8—figure supplement 3), that shows that Foxa2 positive region of the foregut is expanded from head fold to early somite stages. In these mutants this expansion is coupled to a failure in gut folding and the formation of the anterior intestinal portal (Figure 8). Text on these results can be found in the last paragraph of the Results section.

• While we appreciate the reviewers are interested in later foregut phenotype, this is complicated by the severe morphological defects in the global *Fn1-null* and *Itga5-null* mutants at later stages of development (George et al., 1997, [9], Georges-Labouesse et al., 1996, Goh et al., 1997, [48]). We are interested in the role of *Fn1* and Itga5 in endoderm morphogenesis, but this will require conditional mutagenesis, that, although very interesting, is beyond the scope of this manuscript.

*4) The phenotypes of Fn*^*-/-*^
*embryos have been reported to be very variable dependent on genetic background, and so it is important to understand the overall phenotype seen in their particular mutant strain*.

The phenotype of *Fn1-null* embryos varies depending on the genetic background; it is more severe on *129S4* than on *C57BL/6J* background (George et al., 1997). However, we have observed the expansion of Cer1 and the prospective foregut domain on all genetic backgrounds tested; *C57BL/6J* (Figure 8—figure supplement 2) *129S4* (Figure 8 and Figure 8—figure supplement 3) and a *C57/129* hybrid (Figure 8). Expansion of Cer1 and Foxa2 in *Itga5* mutants was done on a *129S4* background. The issue is now mentioned in the Results section.

*5) In the discussion, they state that there is no gastrulation phenotype in Fn mutants, and yet they are actually reporting such a phenotype with the Cer1 expansion. Clearly a further analysis of the effect of the Fn mutation on gastrulation, mesoderm and endoderm formation is called for*.

• We agree that more extensive analysis of the endoderm was warranted. In particular we assessed the expression of Sox17 as we had observed that Sox17 was expressed at higher levels in naïve endoderm produced in response to block PI3K signaling in vitro (Figure 2) and in vivo (Figure 8). Strikingly we observed that Sox17 expression in vivo mirrored *Fn1* and was reduced in the Cer1 prospective foregut as it moved away from the distal tip. In the *Fn1* mutants, Sox17 levels were reduced in the population of cells ectopically expressing Cer1, consistent with a model that high Sox17 levels were supported by PI3K/Fn1 signaling (Figure 8—figure supplement 3). We have also added additional data on Foxa2 (see above). The description of the phenotype is now expanded and covered in the Results.

• In both the *Fn1* and *Itga5* mutants, there are mesoderm defects, in the notochord and somites. However, there appears little change in the specification of mesoderm, including the expression of *Brachyury (T), Sonic Hedgehog, Notch-I, BMP4* and *Mox-I* (Georges-Labousse et al., 1996, Goh et al., 1997, Pulina et al., 2011). We enclose a figure for the reviewers, demonstrating normal expression of *Brachyury (T)* and *Sonic Hedgehog* (Figure 10), but pointing to a problem with axis elongation. As similar observations have been published previously, we have not included this in the paper, but are willing to add this data if requested.Author response image 1.Dorsal view of WT and *Fn1*^*-/-*^ E8.0 embryos after in situ hybridization showing RNA expression for mesoderm markers *Brachyury (T)* and *Sonic Hedgehog*.

• The phenotype of the *Fn1-null* and *Itga5-null* mutants can be summarized as follows. Morphological abnormalities in *Fn1-null* and *Itga5-null* mutants are obvious by E8.0. However, why these mutants appear abnormal is not well understood. These mutants have an obvious axis extension problem and thus appear smaller than their wild-type or heterozygous littermates at E8.0-E9.5 (Georges-Labouesse et al., 1996, Goh et al., 1997). In addition to the mesoderm, we and others have also shown that dorso-ventral and anterior-posterior neural patterning is normal in these mutants, and they produce neural crest derivatives (Goh et al., 1997, Mittal et al., 2010, Pulina et al., 2011). A general conclusion from all these studies has been that Fn1 and Itga5 are not required for migration, specification and differentiation of various mesodermal lineages (Georges-Labouesse et al., 1996, Goh et al., 1997).

• Thus while our study highlights the importance of exploring the earlier phenotypes in these mutants more carefully, a meaningful study would require extensive analysis of both stages and markers, which is beyond the scope of this manuscript.

*6) Despite their emphasis in the results on FN being the key determinant of the patterning of the endorderm, in the discussion they seem less sure and talk about the complexities of the ECM differences. This part needes some clarification*.

We have further extended and hopefully clarified this point in the discussion by suggesting that the other determinants could modulate the effect of Fn1. We hope it is now clear that we were tryinig to highlight that there may be another ECM molecules acting in concert with Fn1, and that we see evidence of this in our mass spec dataset. This is a subject of contuing work on our group. For instance, HSPG’s and Fn1 are intricately related, and HSPG’s has been shown to affect to Fn1 structure reviewed in (Symes et al., 2010).

References

Burtscher I, Lickert H. 2009. Foxa2 regulates polarity and epithelialization in the endoderm germ layer of the mouse embryo.*Development,***136**:1029-38. doi: 10.1242/dev.028415.

Di-Gregorio A, Sancho M, Stuckey DW, Crompton LA, Godwin J, Mishina Y, Rodriguez TA. 2007. BMP signalling inhibits premature neural differentiation in the mouse embryo.*Development,***134**:3359-69. doi: 10.1242/dev.005967.

George EL, Baldwin HS, Hynes RO. 1997. Fibronectins are essential for heart and blood vessel morphogenesis but are dispensable for initial specification of precursor cells.*Blood,***90:**3073-81.

George EL, Georges-Labouesse EN, Patel-King RS, Rayburn H, Hynes RO. 1993. Defects in mesoderm, neural tube and vascular development in mouse embryos lacking fibronectin.*Development,***119**:1079-91.

Georges-Labouesse EN, George EL, Rayburn H, Hynes RO. 1996. Mesodermal development in mouse embryos mutant for fibronectin.*Dev Dyn,***207**:145-56. doi: 10.1002/(SICI)1097-0177(199610)207:2<145::AID-AJA3>3.0.CO;2-H.

Goh KL, Yang JT, Hynes RO. 1997. Mesodermal defects and cranial neural crest apoptosis in alpha5 integrin-null embryos.*Development,***124**:4309-19.

Hansson M, Olesen DR, Peterslund JM, Engberg N, Kahn M, Winzi M, Klein T, Maddox-Hyttel P, Serup P. 2009. A late requirement for Wnt and FGF signaling during activin-induced formation of foregut endoderm from mouse embryonic stem cells.*Dev Biol,***330**:286-304. doi: 10.1016/j.ydbio.2009.03.026.

Khwaja A, Rodriguez-Viciana P, Wennstrom S, Warne PH, Downward J. 1997. Matrix adhesion and Ras transformation both activate a phosphoinositide 3-OH kinase and protein kinase B/Akt cellular survival pathway.*Embo J,***16:**2783-93. doi: 10.1093/emboj/16.10.2783.

Kwon GS, Viotti M, Hadjantonakis AK. 2008. The endoderm of the mouse embryo arises by dynamic widespread intercalation of embryonic and extraembryonic lineages.*Dev Cell,***15**:509-20. doi: 10.1016/j.devcel.2008.07.017.

Mittal A, Pulina M, Hou SY, Astrof S. 2010. Fibronectin and integrin alpha 5 play essential roles in the development of the cardiac neural crest.*Mech Dev,***127**:472-84. doi: 10.1016/j.mod.2010.08.005.

Morrison GM, Oikonomopoulou I, Migueles RP, Soneji S, Livigni A, Enver T, Brickman JM. 2008. Anterior definitive endoderm from ESCs reveals a role for FGF signaling.*Cell Stem Cell,***3**:402-15. doi: 10.1016/j.stem.2008.07.021.

Pulina MV, Hou SY, Mittal A, Julich D, Whittaker CA, Holley SA, Hynes RO, Astrof S. 2011. Essential roles of fibronectin in the development of the left-right embryonic body plan.*Dev Biol,***354**:208-20. doi: 10.1016/j.ydbio.2011.03.026.

Spence JR, Mayhew CN, Rankin SA, Kuhar MF, Vallance JE, Tolle K, Hoskins EE, Kalinichenko VV, Wells SI, Zorn AM, Shroyer NF, Wells JM. 2011. Directed differentiation of human pluripotent stem cells into intestinal tissue in vitro.*Nature,***470**:105-9. doi: 10.1038/nature09691.

Symes K, Smith EM, Mitsi M, Nugent MA. 2010. Sweet cues: How heparan sulfate modification of fibronectin enables growth factor guided migration of embryonic cells.*Cell Adh Migr,***4**:507-10. doi: 10.4161/cam.4.4.12427.

Yang JT, Rayburn H, Hynes RO. 1993. Embryonic mesodermal defects in alpha 5 integrin-deficient mice.*Development,***119**:1093-10.